# Proximity labelling of pro-interleukin-1α reveals evolutionary conserved nuclear interactions

Rose Wellens [1,2,3], Victor S. Tapia [1,2,3], Paula I. Seoane [1,2,3], Hayley Bennett [4], Antony Adamson [4], Graham Coutts[1,2,3], Jack Rivers-Auty [5], Martin Lowe[6], Jack P. Green [1,2,3], Gloria Lopez-Castejon [3,7], David Brough [1,2,3] ✉ & Christopher Hoyle [1,2,3] ✉

Interleukin-1α is a suggested dual-function cytokine that diverged from interleukin-1β in mammals potentially by acquiring additional biological roles that relate to highly conserved regions in the pro-domain of interleukin-1α, including a nuclear localisation sequence and histone acetyltransferase-binding domains. Why evolution modified pro-interleukin-1α's subcellular location and protein interactome, and how this shaped interleukin-1α's intracellular role, is unknown. Here we show that TurboID proximity labelling with pro-interleukin-1α suggests a nuclear role for pro-interleukin-1α that involves interaction with histone acetyltransferases, including EP300. We also identify and validate inactivating mutations in the pro-interleukin-1α nuclear localisation sequence of multiple mammalian species, including toothed whales, castorimorpha and marsupials. However, histone acetyltransferase-binding domains are conserved in those species that have lost pro-interleukin-1α nuclear localisation. Together, these data suggest that histone acetyltransferase binding and nuclear localisation occurred together, and that while some species lost the nuclear localisation sequence in their pro-interleukin-1α, histone acetyltransferase binding ability was maintained. The nuclear localisation sequence was lost from several distinct species at different evolutionary times, suggesting convergent evolution, and that the loss of the nuclear localisation sequence confers some important biological outcome.

Inflammation is typically a beneficial, coordinated response of immune cells that protects a host from pathogenic infection. Communication between immune cells is mainly driven by signalling molecules called cytokines, which are a large group of proteins that includes chemokines, and growth factors. Cytokines are usually secreted proteins that bind to receptors on target cells to elicit a signalling response. However, there are examples where some cytokines appear to have dual functions through an additional intracellular role[1]. One of the most important cytokine families in the host response to infection is the interleukin-1 (IL-1) superfamily, which consists of 11 members of both pro- and anti-inflammatory cytokines. Pro-inflammatory members of the IL-1 superfamily include the well-studied cytokines IL-1α and IL-1β[2]. IL-1β is a protein that first arose >400 million years ago and is conserved throughout vertebrates, while IL-1α arose as a gene duplication of IL-1β between 320 and 180 million years ago in the proto-mammal, and is, therefore, present only in mammals[3].

Like IL-1β, IL-1α is produced as a less active 31 kDa precursor (pro-IL-1α) that consists of a pro-domain and a mature domain, enzymatic cleavage of which produces the mature, fully active form, which signals through the type 1 IL-1 receptor when released. However, the pro-

domains of IL-1α and IL-1β appear to have evolved separate biological roles[3]. The pro-domain of IL-1β in mammals exhibits limited sequence conservation, and it is considered simply to stop the activation and secretion of IL-1β[4]. The pro-domain of IL-1α, on the other hand, is highly conserved amongst mammals and has several domains which may have evolved through a functional specialisation, driving the divergence of IL-1α from IL-1β[3]. Highly conserved sub-domains within the IL-1α pro-domain include a nuclear localisation sequence (NLS), which is a motif of basic amino acids that facilitates trafficking to the nucleus, and two histone acetyltransferase (HAT)-binding domains[3]. Nuclear localisation of pro-IL-1α is observed in both human and mouse cells[5–7]. HATs (also known as lysine acetyltransferases (KATs)) can be located in the nucleus or cytosol, and they mediate the acetylation of lysine residues on proteins, neutralising their positive charge and regulating the function of the protein[8]. Although originally identified for their role in histone modification and chromatin remodelling, HATs have since been characterised for their specialised role in DNA-damage repair, regulating gene expression and the modification of non-histone substrates[9]. Previously published studies have identified a possible role for pro-IL-1α in the regulation of gene expression via binding to HAT complexes[10–12], although the exact nature of its nuclear role remains unclear.

Here we aimed to discover the intracellular function of IL-1α by establishing a pro-IL-1α proximity-based interactome using proximity labelling and by analysing evolutionary evidence. By tagging pro-IL-1α with the enhanced biotin ligase mutant, TurboID[13], we were able to efficiently biotinylate nearby and interacting proteins of pro-IL-1α in cells, revealing a network of potential IL-1α interactors including several HATs. We validated that the toothed whale clade and the rodent suborder castorimorpha, along with several other mammalian species, have lost NLS-dependent nuclear localisation of pro-IL-1α, but maintained HAT-binding domain conservation. Our data suggest the most likely role of intracellular pro-IL-1α is to interact with HATs, and highlights that IL-1α subcellular location has been subjected to evolutionary pressures on several independent occasions.

## Results
### Characterisation of pro-IL-1α proximity labelling
We previously established the highly conserved nature of the pro-domain of pro-IL-1α in mammals in comparison to pro-IL-1β, suggesting additional functions of the pro-domain of IL-1α other than regulating proteolytic cleavage[3]. The pro-domain of pro-IL-1α contains an NLS, which facilitates nuclear import[6]. The majority of mammals have a highly conserved NLS in their pro-domain of pro-IL-1α, suggesting a possible nuclear role. Therefore, we investigated the nuclear interactors of pro-IL-1α by using TurboID proximity labelling to define a pro-IL-1α proximity-based interactome[13]. HeLa cells were transiently transfected to express either human pro-IL-1α, pro-IL-1α-TurboID, or TurboID alone (Fig. 1A). We then characterised the subcellular distribution and expression levels of the constructs using immunocytochemistry. Pro-IL-1α was largely localised to the nucleus (Fig. 1B) consistent with previous reports of ectopically and endogenously expressed IL-1α, and similar to pro-IL-1α expression in primary mouse bone marrow-derived macrophages (Supplementary Fig. 1)[7,14]. The expression pattern of pro-IL-1α-TurboID closely matched that of untagged pro-IL-1α, confirming that the TurboID tag did not affect expression or distribution of the protein (Fig. 1B). When expressed alone, TurboID was diffusely present throughout the entire cell (Fig. 1B). Western blotting was also used to validate protein expression and labelling (Fig. 1C). Addition of biotin (500 μM, 30 min) to transfected cells followed by streptavidin-HRP labelling of either fixed cells, or of protein gels, confirmed the protein expression patterns described above (Fig. 1B) and that proteins within the cell were biotinylated (Fig. 1D). Treatment of pro-IL-1α- and pro-IL-1α-TurboID-expressing cells with the calcium ionophore ionomycin, an established activator

of calpain-dependent pro-IL-1α processing[15], confirmed that the TurboID tag did not interfere with IL-1α processing, or the kinetics of IL-1α processing, suggesting normal trafficking and processing (Fig. 1E). These data confirm that pro-IL-1α-TurboID and TurboID alone were expressed correctly, trafficked normally, and that the biotin ligase activity of the TurboID was functional.

### Pro-IL-1α proximity-based interactome highlights a prominent proximity to HATs
To establish a pro-IL-1α proximity-based interactome, HeLa cells transfected with TurboID alone or pro-IL-1α-TurboID were treated with biotin (500 μM, 30 min), and samples were analysed for biotinylated proteins by mass spectrometry (Fig. 2A). Principal component analysis (PCA) established that the pro-IL-1α-TurboID proximity-based interactome was different from that of TurboID alone (Fig. 2B). Comparison of pro-IL-1α-TurboID and TurboID alone highlighted 56 selectively enriched proteins within the pro-IL-1α proximity-based interactome (Fig. 2C; Supplementary Table 1). Subcellular location analysis using ingenuity pathway analysis (IPA) of significantly enriched proteins indicated that proteins interacting with pro-IL-1α-TurboID were largely associated with the nucleus (Fig. 2D; Supplementary Fig. 2). However, of all the nuclear proteins enriched by pro-IL-1α-TurboID, only 5.5% of these were significantly enriched (Supplementary Fig. 3). Equally, some nuclear proteins were enriched by TurboID alone, 10.9% of which were significantly enriched. This suggested that the pro-IL-1α-interacting proteins identified were specific to pro-IL-1α itself (Supplementary Fig. 3). Further subcellular location analysis of significantly enriched nuclear proteins revealed an enrichment at chromosomes (Fig. 2E). IPA was also used to determine the canonical pathways and biological functions predicted to be associated with the pro-IL-1α proximity-based interactome based on the representation of the 56 significant proteins within these processes (Fig. 2F). Multiple nuclear receptor signalling pathways were significantly enriched, including the aryl hydrocarbon receptor, glucocorticoid receptor, vitamin D receptor, thyroid receptor and retinoic acid receptor signalling pathways. Further to this, gene expression was the most significant biological function associated with the pro-IL-1α proximity-based interactome (Fig. 2G). These data highlight an association between pro-IL-1α and many nuclear proteins involved in the regulation of a broad array of gene expression pathways. Based on a database of the current literature, STRING analysis determined known interacting partners within the pro-IL-1α proximity-based interactome and used Gene Ontology (GO) terms to identify functions of the pro-IL-1α proximity-based interactome proteins. This analysis identified many members of the pro-IL-1α proximity-based interactome as HAT complex proteins (Fig. 3A). The interaction between pro-IL-1α and the HAT protein, EP300, was previously identified[10,14]. Other HAT proteins identified here that have not previously been recorded to interact with pro-IL-1α are CRT2C, NCOA2, NCOA3, ZZZ3, SUPT20H, EPC1, DMAP1, YEATS2 and EP400 (Fig. 3A). The identification of many HAT-interacting proteins within the pro-IL-1α proximity-based interactome is a strong indication of an intracellular role of pro-IL-1α.

In order to validate some of the significantly enriched protein hits from the proximity labelling experiment, we repeated the experiment by transfecting HeLa cells with pro-IL-1α-TurboID and TurboID alone, followed by treatment with biotin. We then performed a streptavidin pull-down of the biotinylated proteins, before analysing the streptavidin-enriched eluates by western blotting (Fig. 3B). Both pro-IL-1α-TurboID and TurboID strongly biotinylated themselves, and therefore were strongly enriched in their respective eluates. The HAT protein EP300 was enriched in the eluate of cells transfected with pro-IL-1α-TurboID compared to TurboID alone, as was ZZZ3, another HAT protein identified in the mass spectrometry analysis (Fig. 3B)[16]. Furthermore, we confirmed that NCAPH, a protein that we had identified in the mass spectrometry analysis as being significantly enriched by

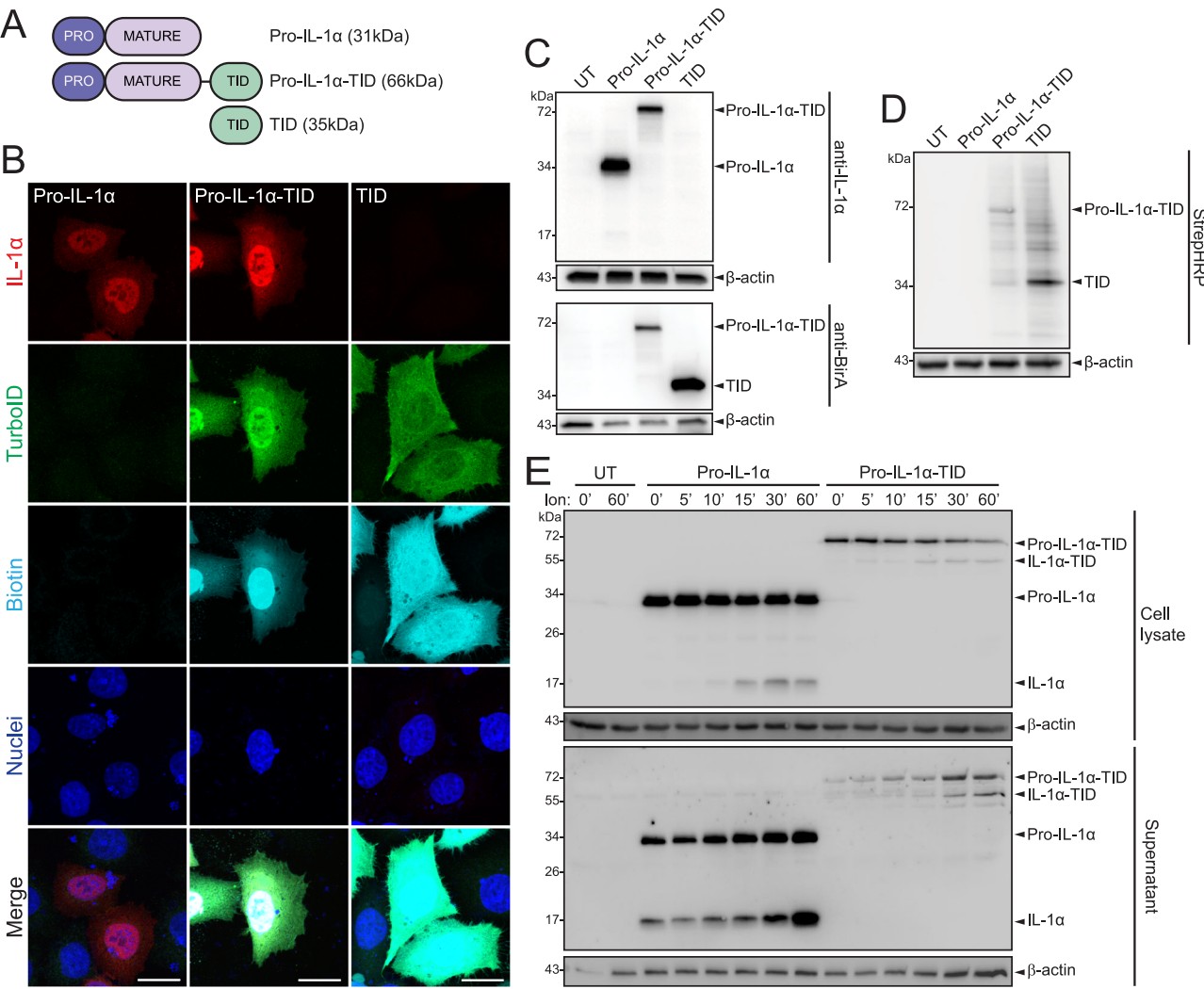

**Fig. 1 | Characterisation of pro-IL-1α-TurboID in HeLa cells. A** Schematic of pro-IL-1α, pro-IL-1α-TurboID (pro-IL-1α-TID), and TurboID (TID) constructs. **B** HeLa cells were transfected with constructs in **A**, then treated with biotin (500 μM, 30 min), and analysed by immunofluorescence microscopy (*n* = 3). Anti-IL-1α labels pro-IL-1α and mature IL-1α, anti-BirA labels TurboID, streptavidin-HRP labels biotinylated proteins. Dark blue represents nuclei stained by DAPI. Scale bars are 20 μm. **C**, **D** HeLa cells were untransfected (UT) or transfected with constructs in **A**, then treated with biotin (500 μM, 30 min). Cell lysates were probed for **C** IL-1α (anti-IL-1α), TurboID (anti-BirA) (*n* = 4), and **D** biotin (StrepHRP) (*n* = 4). **E** HeLa cells were untransfected (UT) or transfected with pro-IL-1α or pro-IL-1α-TID and were then treated with ionomycin (Ion) for 0, 5, 10, 15, 30 or 60 min (*n* = 4). Cell lysates and supernatants were probed for IL-1α by western blotting. 0 and 60 min were used as representatives for UT cells. Source data are provided as a Source Data file.

TurboID (Supplementary Fig. 3C), was enriched in the eluate of cells transfected with TurboID alone (Fig. 3B). Finally, immunofluorescence labelling revealed that both EP300 and ZZZ3 co-localised with pro-IL-1α in the nucleus of HeLa cells (Fig. 3C, D) and in human THP-1 macrophages (Supplementary Fig. 4), indicating that these proteins reside in the same subcellular compartment, with EP300 exhibiting a homogeneous distribution throughout the nucleus, whereas ZZZ3 formed puncta. ANKRD17, a protein that was significantly enriched by TurboID alone (Supplementary Fig. 3C), also exhibited nucleoplasmic and nuclear membrane localisation, and co-localised with pro-IL-1α in the nucleus of HeLa cells (Fig. 3E). Thus, nuclear localisation alone was not sufficient for significantly enriched biotinylation by pro-IL-1α-TID.

### Several mammalian species, including all castorimorpha, lack pro-IL-1α NLS conservation

Although the NLS is well conserved in the pro-IL-1α of most mammals (Fig. 4A), we recently identified that within the order cetacea, the toothed whale clade (odontceti) contained mutations in the pro-IL-1α NLS 'KKRR' motif, resulting in predicted loss of NLS function, whereas the baleen whale clade (mysticeti) retained KKRR motif conservation[3]. Given recent advances in sequence availability, we were able to investigate if pro-IL-1α contained a non-functional NLS in other mammals. We retrieved and aligned 226 pro-IL-1α amino acid sequences from 157 mammalian species, and assessed the conservation of the pro-IL-1α NLS KKRR motif in these species (for full protein sequence alignment see Supplementary Data 1). We identified numerous mammalian species, in addition to toothed whales, which contain amino acid mutations in the pro-IL-1α NLS KKRR motif leading to a predicted loss of NLS function (Fig. 4B, Supplementary Fig. 5A). We also identified the KKRR motif nucleotide sequences for these species (Supplementary Fig. 5B). These species were the naked mole-rat (*Heterocephalus glaber*), southern grasshopper mouse (*Onychomys torridus*), American beaver (*Castor canadensis*), Ord's kangaroo rat (*Dipodomys ordii*), banner-tailed kangaroo rat (*Dipodomys spectabilis*), Chinese pangolin (*Manis pentadactyla*), big brown bat (*Eptesicus fuscus*), Tasmanian devil (*Sarcophilus harrisii*), koala (*Phascolarctos cinereus*) and common brushtail (*Trichosurus vulpecula*). There may be additional mammalian species that also have NLS mutations for which

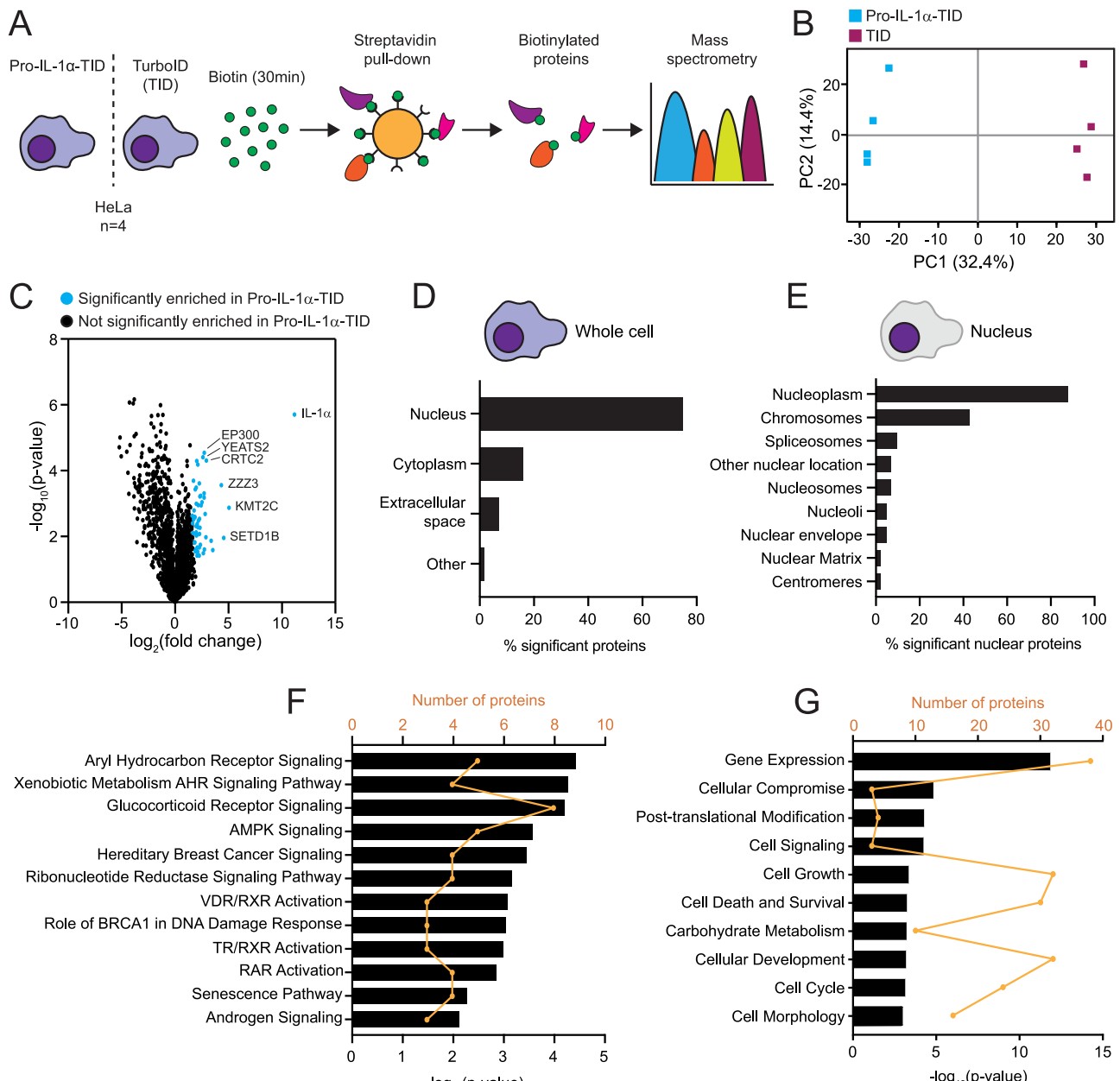

**Fig. 2 | Spatial and functional profiling of pro-IL-1α-TurboID. A** Schematic illustrating the pipeline of biotinylation enrichment analysis by mass spectrometry. HeLa cells were transfected with pro-IL-1α-TurboID or TurboID (TID), and then treated with biotin (500 μM, 30 min, *n* = 4). Streptavidin pull-down of biotinylated proteins was performed, and biotinylated proteins were analysed by mass spectrometry. **B** Principal component analysis of LFQ intensities. A single square denotes an independent experiment. Experimental groups (Pro-IL-1α-TID or TID) are grouped by colour. **C** Volcano plot of proteins enriched in Pro-IL-1α-TID (blue) compared to TID following two-sample two-tailed t-test of log₂-transformed LFQ intensity values (significance determined as s0 = 2; FDR = 0.01) using Perseus

software. **D**–**G** Ingenuity pathway analysis (IPA) of proteins significantly enriched in Pro-IL-1α-TID. **D** Subcellular location of all significantly enriched proteins. **E** Nuclear location of significantly enriched nuclear proteins. **F** Canonical pathways (*p* < 0.01) and **G** biological functions (top 10) associated with proteins significantly enriched in Pro-IL-1α-TID. Black bars represent log₁₀(*p* value) of overlap between Pro-IL-1α-TID proteins and the proteins within each canonical pathway or biological function, and were determined using the right-tailed Fisher's exact test implemented in the IPA software. Orange line denotes number of Pro-IL-1α-TID proteins present within each canonical pathway or biological function. Source data are provided as a Source Data file.

we did not retrieve an IL-1α sequence. We subsequently analysed the divergence times of these species, which suggested that the pro-IL-1α NLS loss events we detected were evolutionary distinct, as they occurred in diverse mammalian lineages (Fig. 4B, Supplementary Fig. 6)[17]. These data suggested that multiple mammalian species may have lost pro-IL-1α nuclear localisation following distinct evolutionary events.

To more precisely identify the evolutionary time-point when the NLS mutations arose in these lineages, and given that the initial

alignment performed did not include sequences from all mammalian species, where possible we retrieved *IL1A* exon 3 (which contains the NLS coding sequence) DNA or RNA sequence reads from species closely related to the identified NLS mutant species (Supplementary Fig. 7A; Supplementary Data 2). This approach also allowed us to retrieve *IL1A* nucleotide sequences from species whose genome or RNA had been sequenced but not necessarily annotated for *IL1A*. We then compared whether the KKRR motif nucleotide sequence region from these reads matched that of the human KKRR motif nucleotide

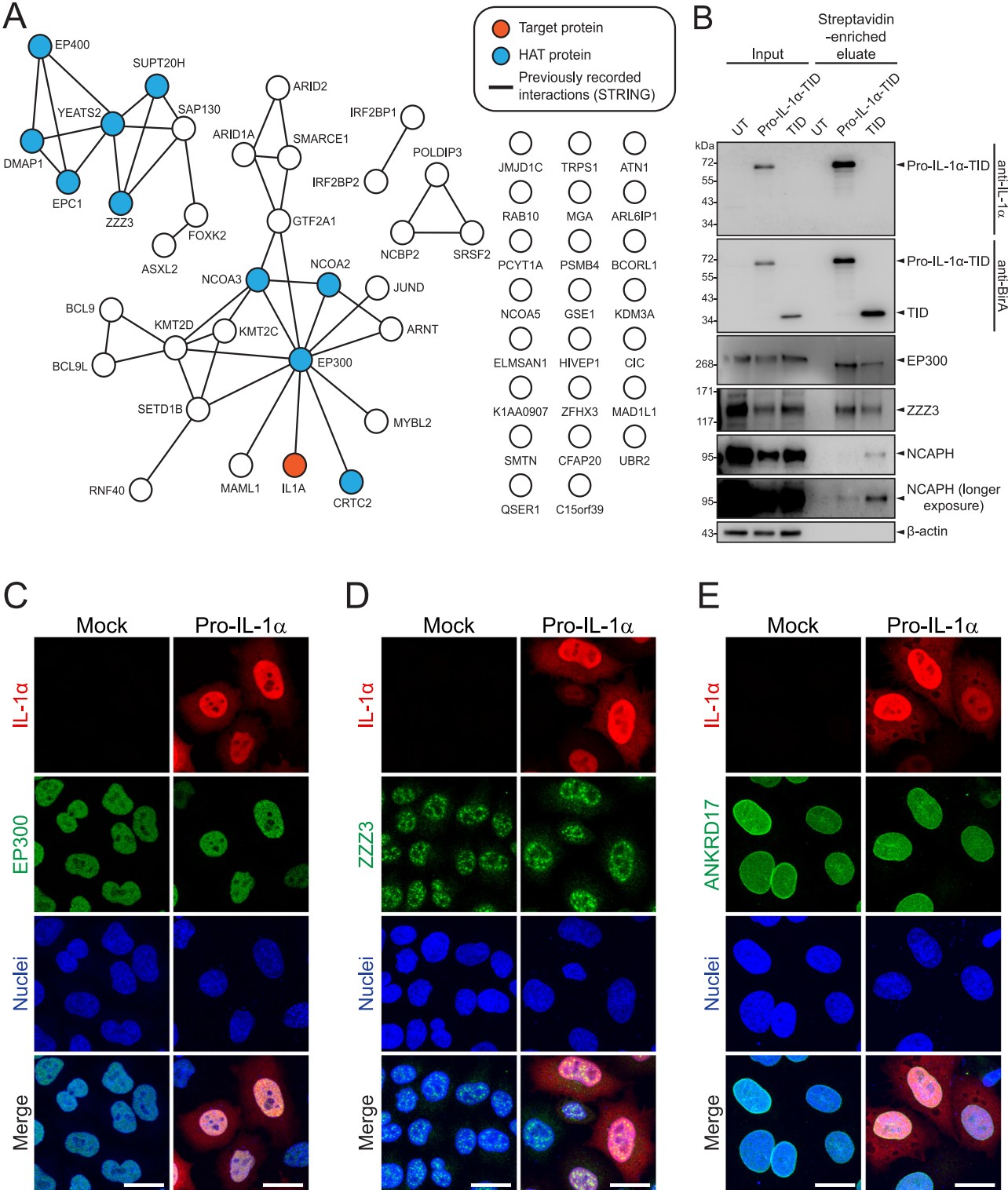

**Fig. 3 | STRING analysis of pro-IL-1α-TurboID biotinylated proteins highlighting prominent proximity to histone acetyltransferase (HAT) proteins.**
**A** STRING network analysis (generated via https://string-db.org/) of proteins significantly enriched in pro-IL-1α-TurboID. Lines represent previously recorded physical interactions identified by STRING, unconnected proteins have no previously recorded interactions with significantly enriched proteins. Histone acetyltransferase (HAT) proteins were identified through gene ontology (GO) analysis. IL1A (pro-IL-1α) represents the target protein of this experiment. **B** HeLa cells were untransfected (UT) or transfected with pro-IL-1α or pro-IL-1α-TID, and then treated with biotin (500 μM, 1 h). Cell lysates (input) and streptavidin-enriched eluates were probed by western blotting for IL-1α (anti-IL-1α), TurboID (anti-BirA), EP300, ZZZ3 or NCAPH (*n* = 4). **C**–**E** HeLa cells were untransfected (mock) or transfected with pro-IL-1α, and analysed by fluorescence microscopy for pro-IL-1α and (**C**) EP300, (**D**) ZZZ3 or (**E**) ANKRD17 (*n* = 4). Representative maximum projection confocal immunofluorescence images are shown. Scale bars are 20 μm. Source data are provided as a Source Data file.

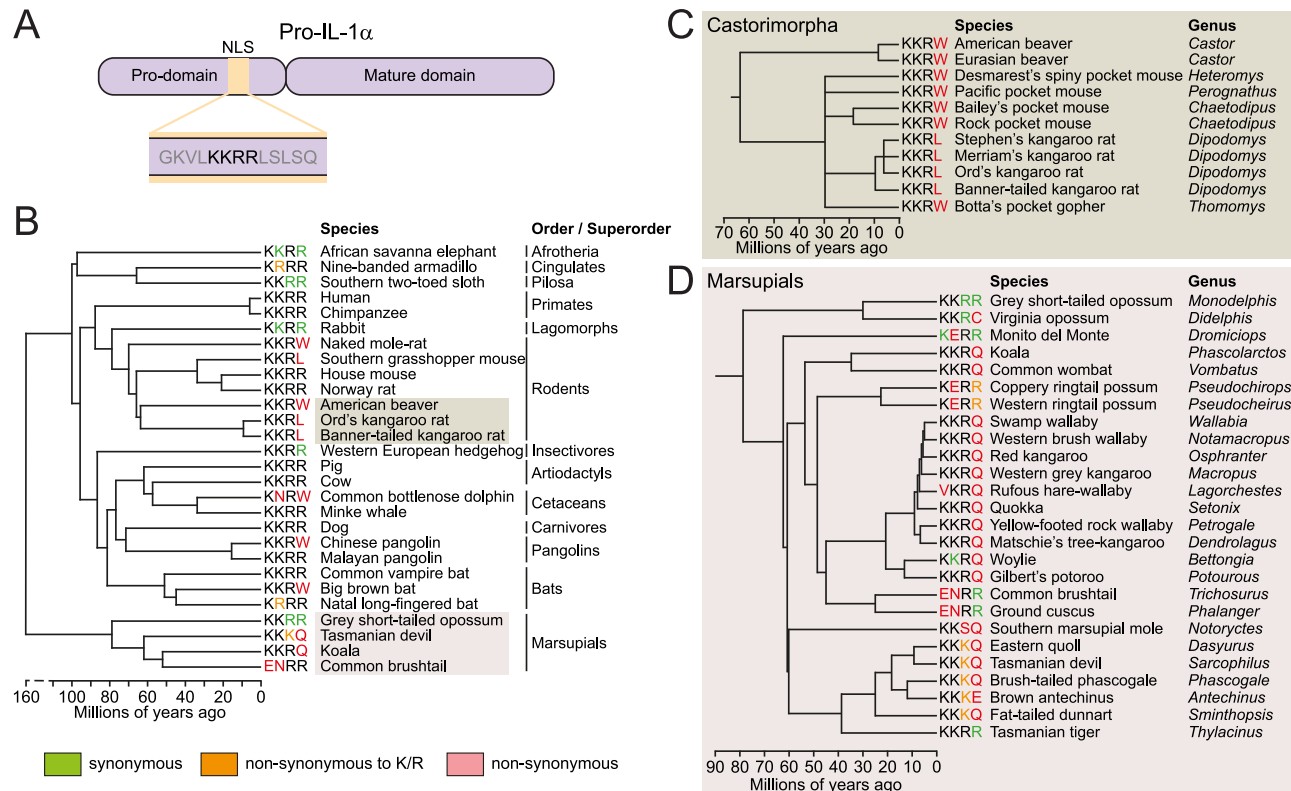

**Fig. 4 | Several mammalian species lack pro-IL-1α nuclear localisation sequence (NLS) conservation. A** Schematic of pro-IL-1α protein domains, including the NLS location and amino acid sequence. **B** Model mammalian evolutionary tree, highlighting representative mammalian species from various orders and superorders, as well as the identified species with NLS mutations in the KKRR motif. KKRR motif sequences were retrieved from the full IL-1α protein alignment from all isoforms of all 157 species (see Supplementary Data 1). **C** An expanded model tree of the rodent suborder castorimorpha, and **D** an expanded model tree of the marsupial lineage, only including species for which an *IL1A* sequence could be retrieved from the sequence read analysis (see Supplementary Data 2). Non-synonymous mutations are shown in red, apart from mutations to K/R, which are shown in orange. All model trees were separately manually generated using divergence times retrieved from TimeTree[17].

sequence, which was conserved across many mammalian species (Supplementary Fig. 5B). Sequence reads from the NLS mutant species themselves were also assessed to validate the presence of the mutated KKRR motif sequences. We identified that, similar to the toothed whale clade, all species of the rodent suborder castorimorpha, which contains beavers (*Castoridae*), pocket mice and kangaroo rats (*Heteromyidae*) and gophers (*Geomyidae*), for which we could obtain sequence reads possessed non-synonymous nucleotide mutations in the KKRR motif, resulting in an amino acid substitution of the final arginine residue to tryptophan or leucine (Fig. 4C, Supplementary Fig. 7B, C). These data suggested that this NLS mutation arose after the castorimorpha lineage diverged from its rodent ancestors ~65–70 million years ago[17–19], although other estimates place this divergence later at ~50–55 million years ago[20], and it was then subsequently conserved. Most of the castorimorpha species possessed a KKRW motif, but the kangaroo rat sequences analysed contained a KKRL motif, caused by an additional non-synonymous nucleotide mutation in the final amino acid residue that occurred in kangaroo rats after they further diverged (Fig. 4C, Supplementary Fig. 7B). Through this sequence read analysis, we were able to validate the loss of the NLS in the naked mole-rat, southern grasshopper mouse, Chinese pangolin and big brown bat, but the close relatives of these species that we assessed all possessed an intact NLS, with no apparent non-synonymous nucleotide mutations in the KKRR motifs (Supplementary Figs. 8–11). Thus, the NLS loss was specific to these individual mutant NLS species, and these were isolated, separate evolutionary events. The marsupial lineage contained multiple species with varying non-synonymous nucleotide mutations in the NLS KKRR motif,

indicating an almost global lack of NLS conservation in this mammalian group (Fig. 4D, Supplementary Figs. 12 and 13).

## KKRR motif mutations result in reduced pro-IL-1α nuclear localisation and enhanced release of mature IL-1α

We had identified species that possessed mutations in the pro-IL-1α NLS sequence, resulting in predicted loss of pro-IL-1α nuclear localisation, but we needed to validate this experimentally. Therefore, we transfected HeLa cells with either full length human pro-IL-1α, full length human pro-IL-1β which does not have an NLS and therefore is localised to the cytosol, a chimeric IL-1 protein composed of a human IL-1α pro-domain fused to the human IL-1β mature domain (pro-α-mat-β), or a chimeric IL-1α protein composed of a toothed whale (*Orcinus orca*, killer whale) IL-1α pro-domain fused to the human IL-1α mature domain (Orca pro-IL-1α) (Fig. 5A), and then assessed cytokine subcellular localisation. Full length pro-IL-1α was predominantly located in the nucleus of the cells, whereas full length pro-IL-1β was largely cytosolic (Fig. 5B, C). Fusing the IL-1α pro-domain to the IL-1β mature domain resulted in nuclear localisation of this protein, confirming that nuclear localisation is an inherent phenotype driven by the IL-1α pro-domain (Fig. 5B, C). The chimeric Orca pro-IL-1α was distributed throughout the cell cytoplasm, confirming that the Orca pro-domain of pro-IL-1α does not contain a functional NLS, and therefore that toothed whale pro-IL-1α is not trafficked to the nucleus, but instead is predominantly located in the cytosol (Fig. 5B, C).

It was possible that the reduction in nuclear localisation caused by the Orca IL-1α pro-domain may have been due to differences in the rest of the pro-IL-1α NLS sequence, and not specifically due to the

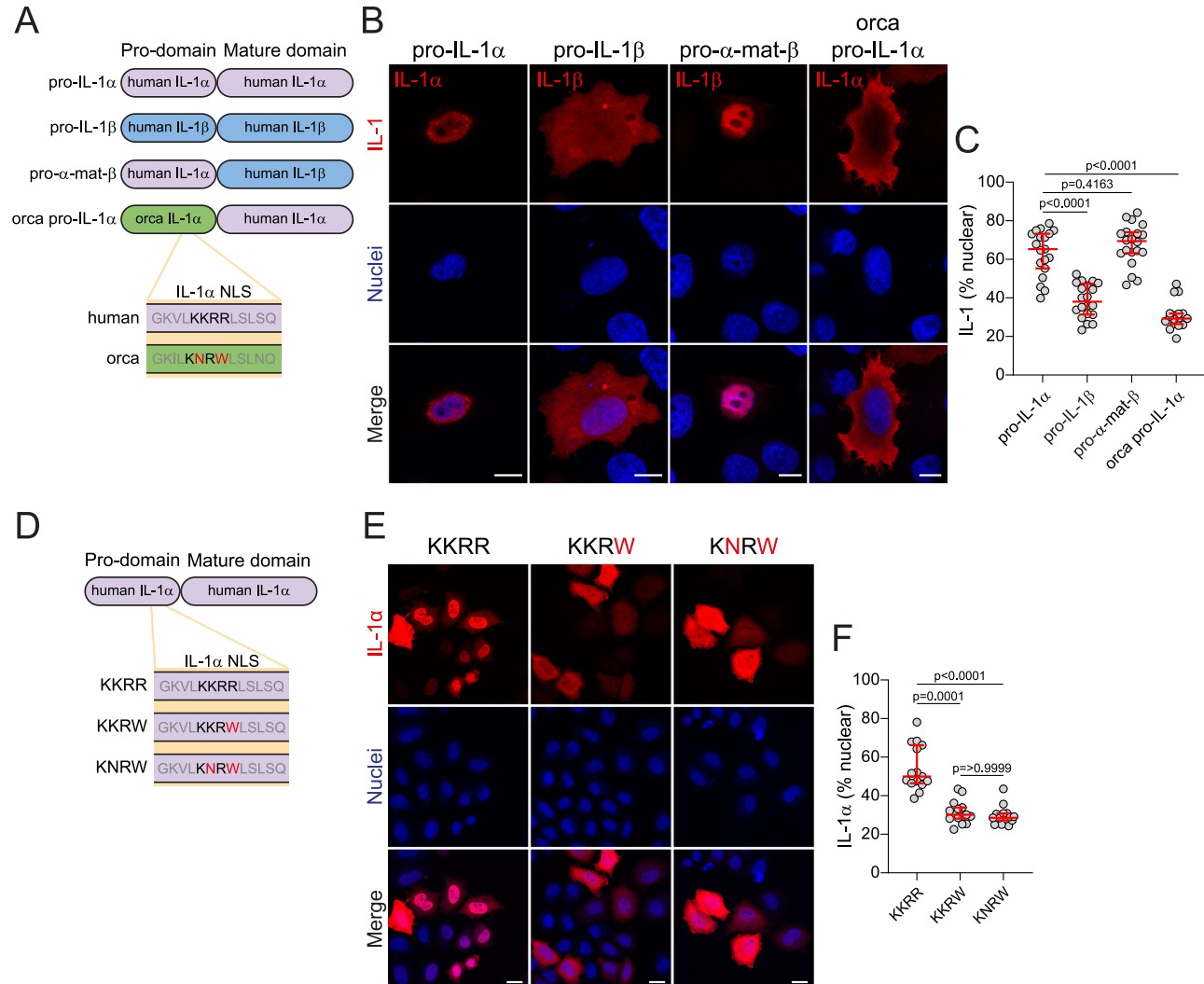

**Fig. 5 | KKRR motif mutations in the nuclear localisation sequence (NLS) reduce pro-IL-1α nuclear localisation. A–C** HeLa cells were transfected with full length human pro-IL-1α, full length human pro-IL-1β, human IL-1α pro-domain fused to the human IL-1β mature domain (pro-α-mat-β), or Orca IL-1α pro-domain fused to the human IL-1α mature domain (Orca pro-IL-1α). **A** Schematic of constructs. **B** Representative immunofluorescence images and **C** quantification of IL-1 nuclear localisation are shown ($n$ = 19 (pro-IL-1α), $n$ = 20 (pro-IL-1β), $n$ = 20 (pro-α-mat-β), $n$ = 18 (orca pro-IL-1α) fields of view from four independent experiments). Scale

bars are 10 μm. **D–F** HeLa cells were transfected with human pro-IL-1α plasmids containing a KKRR, KKRW, or KNRW NLS motif. **D** Schematic of constructs. **E** Representative maximum projection confocal immunofluorescence images and **F** quantification of IL-1α nuclear localisation are shown ($n$ = 15 fields of view from three independent experiments). Scale bars are 20 μm. Data are median ± IQR. Data were analysed using one-way ANOVA followed by Dunnett's post-hoc test (**C**) or Kruskal–Wallis test followed by Dunn's post-hoc test (**F**). Source data are provided as a Source Data file.

mutations in the KKRR motif that resulted in a KNRW motif instead. To determine whether mutations in the KKRR motif alone were sufficient to cause a reduction in nuclear localisation, we transfected HeLa cells with either human WT pro-IL-1α (KKRR) or a mutated human pro-IL-1α containing the toothed whale KNRW motif within its NLS (Fig. 5D). WT pro-IL-1α was strongly localised to the nucleus, whereas introduction of the double amino acid mutation (KNRW) from toothed whales into the pro-IL-1α construct reduced pro-IL-1α nuclear localisation, and enhanced its cytosolic distribution, indicating that these mutations alone were sufficient for loss of NLS function (Fig. 5E, F). Whilst the toothed whale NLS sequences typically contained two amino acid mutations in the KKRR motif, resulting in KNRW, the newly identified NLS mutant species typically only contained a mutation in the final arginine residue of this motif, most commonly to KKRW, which incidentally is the same sequence as the sperm whale, another member of the toothed whale clade (Fig. 5D). Thus, to establish whether a single amino acid mutation in the KKRR motif was sufficient to cause a reduction in pro-IL-1α nuclear localisation, we also transfected HeLa

cells with human pro-IL-1α containing a KKRW motif within its NLS, and this too reduced pro-IL-1α nuclear localisation, indicating loss of NLS function (Fig. 5E, F). The loss of pro-IL-1α nuclear localisation caused by the NLS mutations also resulted in reduced co-localisation with EP300 (Supplementary Fig. 14). These data suggest that the loss of KKRR motif conservation in the newly identified NLS mutant species, such as castorimorpha and marsupials, would result in reductions in pro-IL-1α nuclear localisation, similar to toothed whales, and that this may reduce interactions with nuclear HAT proteins.

We also tested whether reduced pro-IL-1α nuclear localisation would affect the cleavage and release of IL-1α. HeLa cells were transfected with human pro-IL-1α or the Orca IL-1α pro-domain fused to the human IL-1α mature domain and subsequently treated with ionomycin (as above) for one or four hours. Release of Orca pro-IL-1α into the supernatant was not significantly increased after one hour of ionomycin treatment, but we observed increased cleavage of Orca pro-IL-1α into its mature form, with no difference in ionomycin-induced cell death (Supplementary Fig. 15A–E). Following four hours of ionomycin

treatment, the release of Orca pro-IL-1α into the supernatant was enhanced, again with no difference in ionomycin-induced cell death (Supplementary Fig. 15F–H). We also transfected HeLa cells with the human pro-IL-1α plasmids containing a WT (KKRR), KNRW, or KKRW NLS motif, and treated these cells with ionomycin for one hour. The NLS mutations resulted in enhanced IL-1α release in response to ionomycin, with no difference in ionomycin-induced cell death (Supplementary Fig. 15I–K), as well as enhanced cleavage of pro-IL-1α in the supernatant and lysate (Supplementary Fig. 15L–N).

### HAT-binding domains are conserved in most mammalian species, including pro-IL-1α NLS mutants

Binding motifs for HATs have previously been described in the pro-domain of pro-IL-1α, and HATs have been described to interact with pro-IL-1α[10]. Having validated the loss of pro-IL-1α NLS function in numerous mammalian species, this meant we could further explore the evolutionary relationship between pro-IL-1α nuclear localisation and HAT binding. To do this, we assessed the conservation of these HAT-binding domains in the mammalian species with an intact NLS and compared this to species with a mutated KKRR motif sequence. Conservation of the HAT-binding domains appeared consistent between the intact and mutant NLS species (Fig. 6A, B). The N-terminal HAT-binding domain (amino acid residues 7–19) was much better conserved in comparison to the other HAT-binding domain (amino acid residues 98-108) (Fig. 6B). There was no clear difference in conservation of the whole pro-domain between species with an intact NLS and NLS mutants (Fig. 6C). Marsupials had moderate conservation of the N-terminal HAT-binding domain, and poor conservation of the second, although these species had poor conservation across the whole IL-1α pro-domain (Fig. 6A–C) and mature domain compared to other species (Supplementary Fig. 16). This poor sequence conservation in marsupials may also explain why pro-IL-1α NLS mutations were detected in most marsupial species (Fig. 4D), and why marsupials have poor conservation of the thrombin cleavage site in IL-1α[21]. Thus, the species in which the NLS was mutated retained similar conservation of HAT-binding domains in the absence of an intact pro-IL-1α NLS, potentially suggesting that the intracellular role of pro-IL-1α was driven by HAT binding, and not by nuclear localisation solely due to the NLS. This suggests that the function of the pro-IL-1α NLS remains unclear and may have evolved to facilitate interactions with nuclear HATs, or for another reason entirely.

### Monotremes have an intact NLS in pro-IL-1α, and evidence of HAT-binding domain conservation

To further dissect the temporal origin of the NLS and HAT-binding regions, we analysed the *IL1A* gene region in monotremes, which diverged from the mammalian lineage approximately 180–185 million years ago (20–25 million years before marsupials), and whose genomes have recently been sequenced[22] (Fig. 7A). We located the *IL1A* gene in the platypus (*Ornithorhynchus anatinus*) and short-beaked echidna (*Tachyglossus aculeatus*) genome assemblies, although this was unannotated in the platypus, and observed similar chromosomal anatomy to other mammals, although the *CENPT* gene was located between *IL1B* and *IL1A* in both monotreme species (Fig. 7B). We next determined the IL-1α amino acid and nucleotide sequences for these species by locating the *IL1A* exons, and identified a relatively well conserved NLS sequence that was predicted to be functional, and that had a fully conserved KKRR motif (Fig. 7C, D, Supplementary Fig. 17). Despite this conserved NLS, both monotreme pro-IL-1α sequences were relatively poorly conserved compared to the modal pro-IL-1α amino acid sequence determined from the full alignment of all the mammalian species analysed, similar to the marsupials (Fig. 7E, F). It was unsurprising that monotreme and marsupial IL-1α were considerably different from placental mammals, as these mammalian groups diverged from placental mammals more than 150 million years

ago, and it is possible that the marsupial and monotreme sequences may more closely resemble the ancestral IL-1α protein.

We then examined the HAT-binding regions of monotreme pro-IL-1α to see if they were also conserved. The N-terminal HAT-binding domain was moderately conserved in comparison to the other HAT-binding domain, which was poorly conserved (Fig. 7E). We also identified regions in the IL-1α pro-domain (amino acids 1–6 and 32–36) that were highly conserved across monotremes, marsupials and placental mammals, suggesting that these regions may be of fundamental importance for pro-IL-1α function. Thus, these data would place the evolution of the NLS and possibly the N-terminal HAT-binding domain after the *IL-1* gene duplication event in the proto-mammal but before mammalian divergence, driving IL-1α's divergence from IL-1β, and the NLS was subsequently lost in certain mammals in distinct evolutionary events. However, from these data we are unable to conclude whether the NLS or HAT-binding domain appeared first.

## Discussion

IL-1α is a pro-inflammatory cytokine that has been considered to act as a damage-associated molecular pattern (DAMP), or an alarmin, upon its release from damaged cells, and it is implicated in the inflammation contributing to many conditions, including respiratory, cardiovascular, and neurological diseases[23,24]. IL-1α signalling is also suggested to contribute to other inflammatory processes, including cellular senescence where it promotes the senescence-associated secretory phenotype (SASP) (e.g.,[25,26]). However, pro-IL-1α is a putative dual-function cytokine, as it is also characterised by a nuclear subcellular localisation and potential function[1]. Nuclear localisation of pro-IL-1α is driven by an NLS within its pro-domain[6,7]. Our lab previously reported that, whilst the NLS within the pro-domain of pro-IL-1α is conserved amongst many mammalian sequences, it is lost in the toothed whale clade[3]. We previously used this to test the hypothesis that nuclear functionalisation drove the evolution of IL-1α[3]. We predicted that if the NLS was important, then there would be an increase in the amino acid replacements in the pro-domain of toothed whale IL-1α, as without an NLS, the nuclear function would now be redundant. However, the sequence of the pro-domain of toothed whale IL-1α remains highly conserved, despite the functional mature domain of IL-1α acquiring an increase of amino acid replacements. This suggests that a nuclear functionalisation related specifically to the NLS was probably not a major force driving the divergence of IL-1α[3]. However, as we present in this study, even in pro-IL-1α sequences from species where there is no functional NLS, there are conserved HAT-binding domains. This, and the proximity-based interactome described above, suggest that HAT binding may be the primary intracellular role of pro-IL-1α, and that nuclear trafficking driven by the NLS served some auxillary role. This purpose of the NLS may be simply to facilitate binding to HATs that reside in the nucleus, or it may be completely independent of HAT binding, and instead related to the trafficking and release of IL-1α.

We identified nine histone-modifying enzymes as part of the pro-IL-1α proximity-based interactome, in addition to the HAT protein EP300, which is reported to functionally interact with pro-IL-1α in the nucleus to influence EP300-mediated transcriptional activity[10,12,14]. Deletion of the HAT-binding domain regions in pro-IL-1α abolished the interaction between pro-IL-1α and EP300 and reduced this transcriptional activity[10]. Histone modification is a critical process in the regulation of gene expression, which is a key biological function associated with the pro-IL-1α proximity-based interactome identified in this study. This suggests that pro-IL-1α may play an important role in regulating gene expression via its association with proteins involved in histone modification, further supporting its role as a dual-functioning protein.

The mutations causing a loss of the NLS arose at distinct, independent time points in mammalian evolution. Castorimorpha lost NLS function after divergence from the rodent lineage approximately 67

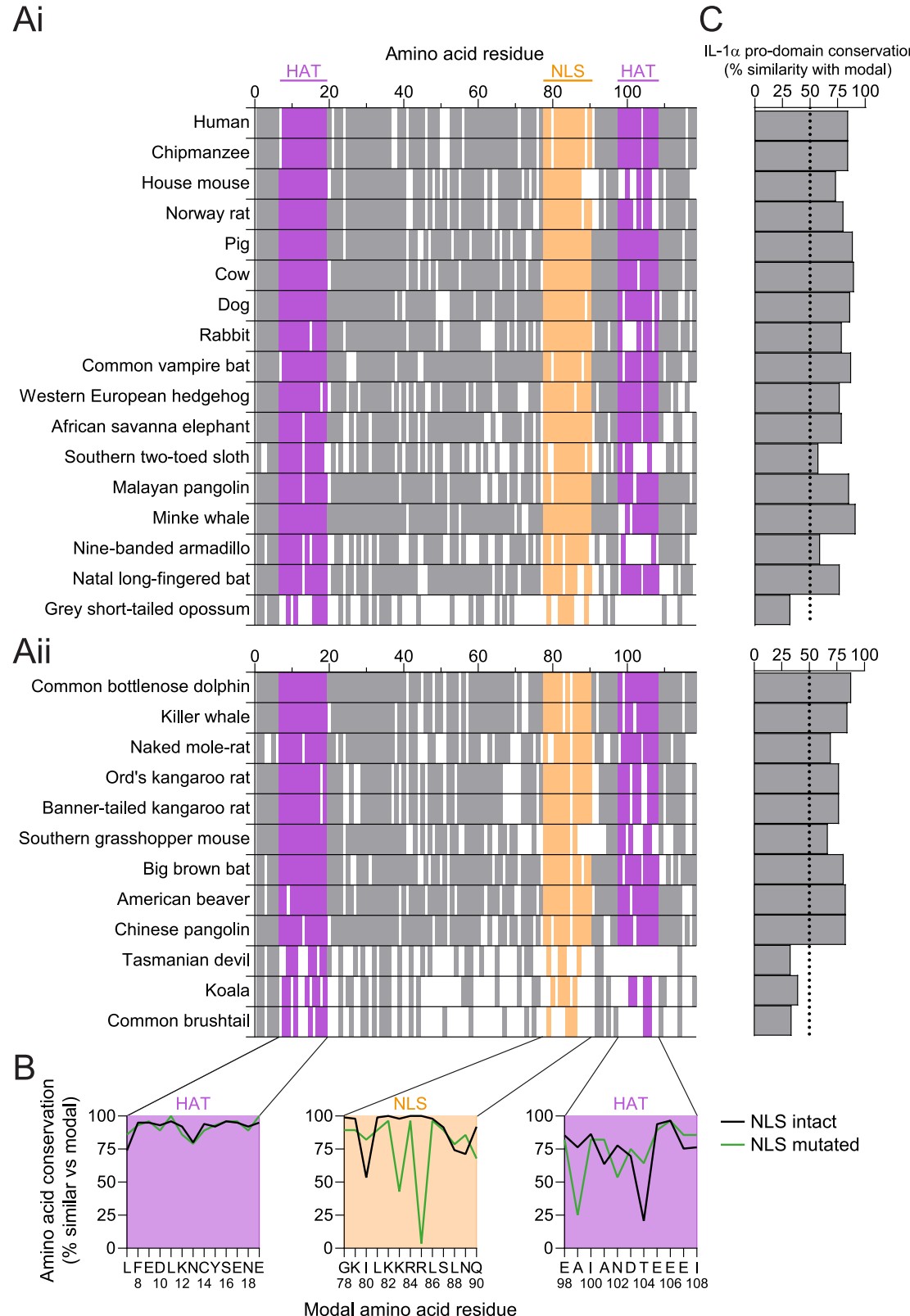

**Fig. 6 | Histone acetyltransferase (HAT)-binding domains are similarly conserved in most mammalian species, including nuclear localisation sequence (NLS) mutants. A** Pro-IL-1α pro-domain amino acid conservation in representative intact NLS (Ai) and mutated NLS (Aii) species. Residues that matched the modal pro-IL-1α amino acid sequence (determined from the full alignment of all the mammalian species analysed−see Supplementary Data 1) are indicated by a solid colour, whereas residues that did not match are indicated by a white gap. Positions where gaps were the modal residue were removed from the alignment. HAT-binding domains are identified in purple (amino acids 7−19, 98−108 in human sequence), and the NLS is identified in orange (amino acids 78−90 in human sequence). **B** Amino acid sequence conservation of both HAT-binding domains and the NLS for all sequences with an intact or mutated NLS (relative to the modal amino acid residue from the full alignment of all sequences. See Supplementary Data 1). **C** Conservation of the whole pro-IL-1α pro-domain relative to the modal pro-domain sequence. Source data are provided as a Source Data file.

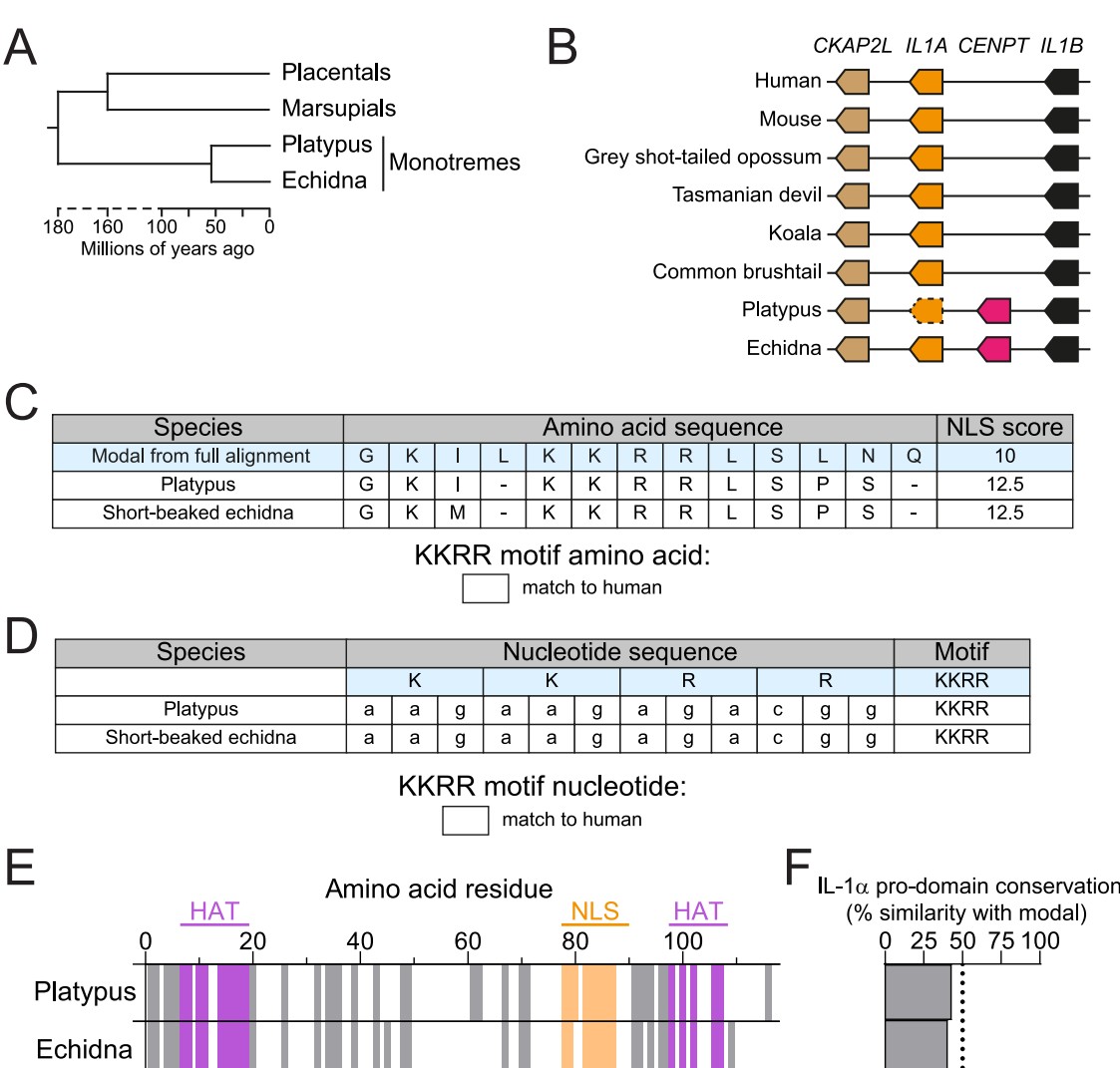

**Fig. 7 | Monotremes have an intact nuclear localisation sequence (NLS) in pro-IL-1α, and evidence of histone acetyltransferase (HAT)-binding domains.**
**A** Model evolutionary tree of the divergence of placental mammals, marsupials and monotremes, generated manually using divergence times retrieved from TimeTree[17]. **B** Chromosomal anatomy of *IL1A* gene region in several mammalian species. Arrow direction indicates gene orientation. Dotted outline indicates lack of genome annotation. Gene lengths and distances between genes are not to scale. **C** Pro-IL-1α NLS amino acid sequences of the platypus and short-beaked echidna. Modal sequence was taken from the alignment of IL-1α from all isoforms of all 157 species. NLS scores were estimated using NLS mapper. Sequences correspond to $G^{78}$-$Q^{90}$ in human sequence. See Supplementary Data 1. **D** KKRR motif nucleotide sequences for the two monotreme species. **E** Pro-IL-1α pro-domain amino acid conservation in the platypus and short-beaked echidna. Residues that match the modal amino acid from the full sequence alignment are indicated by a solid colour, whereas residues that did not match are indicated by a white gap. Positions where gaps were the modal residue were removed from the alignment. HAT-binding domains are identified in purple (amino acids 7–19, 98–108 in human sequence), and the NLS is identified in orange (amino acids 78–90 in human sequence). **F** Conservation of the whole pro-IL-1α pro-domain relative to the modal pro-domain sequence. Source data are provided as a Source Data file.

million years ago[17–19], which would place this shortly before the cretaceous-paleogene mass extinction event that resulted in the loss of 75% of all species, including non-avian dinosaurs. The precise timings of mammalian diversification around this extinction event are unclear[27], however, with the divergence of castorimorpha also estimated between 50–55 million years ago[20]. In toothed whales, the loss of the NLS occurred ~35 million years ago when they diverged from baleen whales[17]. The loss of KKRR motif conservation and predicted NLS function in marsupials may be related to the geographical isolation of Australasian and South American marsupials upon separation of the supercontinent Gondwana, with Australasian marsupials such as the koala and Tasmanian devil losing KKRR motif conservation, whereas the South American opossums retained it. The monito del monte, although an inhabitant of South America, has been taxonomically classified as a closely related sister group of Australasian marsupials that likely diverged shortly before these species migrated

to Australia, possibly explaining why this species has also lost predicted NLS function[28].

The evolutionary convergence on loss of NLS function, not just in toothed whales but in the rodent suborder castorimorpha, marsupials and other mammalian lineages, suggests that modifying the subcellular location of pro-IL-1α may have conferred an evolutionary advantage to these species. However, the events that drove selection for NLS loss are unclear, as are any possible benefits conferred by this mutation, and they may be due to reducing the nuclear function of IL-1α. It is also possible that the NLS serves as a mechanism to regulate the secretion of IL-1α by limiting available pro-IL-1α for processing and secretion by virtue of its subcellular compartmentalisation[14,29]. Thus, a reduction of IL-1α nuclear localisation may enhance its extracellular release. Better understanding the implications of losing IL-1α nuclear localisation in these species may reveal valuable insights into IL-1α biology. Although to our knowledge there is no current evidence

regarding pro-IL-1α nuclear localisation in cells isolated from the NLS mutant species identified in this study, cell lines have been previously derived from various cetacean and marsupial species, including toothed whales, and primary cells have also been cultured[30–33]. Thus, these cells could potentially be used to investigate pro-IL-1α nuclear localisation and function, although myeloid cell lines from these species may offer more relevance for IL-1α expression and function.

Critically, in all these independent mutation events in which NLS functionality was lost, at least one of the pro-IL-1α HAT-binding domains remained highly conserved, suggesting that HAT binding is still applying purifying selection pressure. This could be explained by the NLS mutations preventing active trafficking of pro-IL-1α into the nucleus, but not preventing its passive diffusion. Thus, the strongly reduced, but not completely absent, nuclear localisation of pro-IL-1α may have still been sufficient to retain HAT-binding conservation in the mutant NLS species. Another possibility is that cytosolic pro-IL-1α can interact with cytosolic HATs (or KATs) in the mutant NLS species. The functions of cytosolic HATs can be independent of their acetylation activity, for example the multifunctional EP300 HAT protein is also reported to exhibit E3 and E4 ubiquitin ligase activity[34]. Interestingly, this ligase activity is only functional when EP300 is in the cytosolic fraction of the cell[35]. Pro-IL-1α has been identified to be poly-ubiquitinated and this promotes its processing into the inflammatory mature form, but the ligases involved in the process have not yet been identified[36]. Here we have shown that pro-IL-1α interacts with EP300 and at least two other E3 ligases proteins (RNF40 and UBR2, Supplementary Table 1). Nevertheless, while cytosolic interactions of pro-IL-1α and HAT enzymes may be important, the vast majority of mammalian species have a highly conserved NLS and HAT-binding domains. Furthermore, we have shown here that HAT enzymes and other gene regulatory proteins such as transcription factors are the most enriched group of proteins in the proximity-based interactome of pro-IL-1α, suggesting nuclear HAT interaction to be the most likely deciding factor in pro-IL-1α evolution. We observed that the second HAT-binding domain (amino acid residues 98-108) was generally more poorly conserved than the N-terminal HAT-binding domain. It is currently unclear what level of sequence or structural conservation is required to retain HAT-binding capabilities, and whether there are specific conserved motifs within the HAT-binding domain that are critical for HAT binding. It has been shown that deletion of either HAT-binding domain region alone is sufficient to reduce the functional interaction between pro-IL-1α and EP300, suggesting that both domains are required for this interaction[10].

Thus, IL-1α is likely to be a dual-function cytokine, with an intracellular role dictated by a conserved pro-domain and HAT binding, and an extracellular pro-inflammatory role executed by the mature domain. The loss of the NLS from different mammalian species at distinct evolutionary times, suggesting convergent evolution, indicates an important role for pro-IL-1α subcellular location. Understanding the intracellular role of pro-IL-1α, its interactions with HATs and the consequences of this to a cell and organism may represent the next challenge in understanding IL-1α.

## Methods

All mouse procedures were performed according to the UK Home Office Animals (Scientific Procedures) Act (1986) and were approved by the local Animal Ethical Review Group, The University of Manchester, with appropriate personal and project licences in place. Mice were housed in groups of 2–5 at 21 °C, 65% humidity, with a regulated 12 h light-dark cycle, and were allowed free access to food and water.

### Reagents

Goat anti-human IL-1α (AF-200-NA), goat anti-mouse IL-1α (AF-400-NA), and human IL-1α DuoSet ELISA (DY200) were from R&D Systems. Rabbit anti-human BirA (11582-RP01) was from Sino Biological. Bovine serum albumin (A9647), Anti-β-actin-peroxidase (A3854) and biotin (B4639) were from Sigma-Aldrich. Lipofectamine 3000, rabbit anti-human ZZZ3 (PA5-90723), streptavidin-HRP conjugate (S911), Alexa Fluor™ 488 donkey anti-goat IgG (A-11055), Alexa Fluor™ 594 donkey anti-goat IgG (A-11058), Streptavidin Alexa Fluor™ 594 conjugate (S11227) or Alexa Fluor 647™ donkey anti-rabbit IgG (A-31573), DAPI (D1306) and ProLong Gold antifade mounting reagent (P36934) were from Invitrogen. Rabbit anti-human EP300 (ab275378) was from Abcam. Rabbit anti-human NCAPH (11515-1-AP) was from Proteintech. Cytiva Amersham ECL Prime Western Blotting Detection Reagent (RPN2236) was from GE Healthcare. Protease inhibitor cocktail (539131) was from Merck Millipore. Fugene HD (E2311) and CytoTox 96 nonradioactive cytotoxicity assay (G1780) were from Promega. Rabbit anti-goat IgG (P044901-2) or goat anti-rabbit IgG (P044801-2) secondary antibodies were from Agilent. MagReSyn streptavidin beads (MR-STV010) were from ReSyn Biosciences. Ionomycin calcium salt (CAY11932) was from Cayman Chemical. Rabbit anti-human ANKRD17 (HPA063731) was from Atlas Antibodies.

### Cell culture

HeLa cells (ATCC) were cultured in Dulbecco's modified Eagle's medium (DMEM) containing 10% fetal bovine serum (FBS) and 100 U ml$^{-1}$ penicillin and 100 µg ml$^{-1}$ streptomycin, and were incubated at 37 °C and 5% $CO_2$. Before experiments, cells were seeded at densities of $5 \times 10^4$ cells ml$^{-1}$ or $1 \times 10^5$ cells ml$^{-1}$ in 24-well plates for immunocytochemistry experiments, or $2 \times 10^5$ cells ml$^{-1}$ in 24-well or 10 cm culture dishes for western blotting and proximity labelling experiments, respectively, and left overnight.

THP-1 cells (ATCC) were cultured in RPMI 1640 containing 10% FBS and 100 U ml$^{-1}$ penicillin and 100 µg ml$^{-1}$ streptomycin at 37 °C and 5% $CO_2$. Before experiments, cells were seeded at a density of $1 \times 10^6$ cells ml$^{-1}$ and differentiated using phorbol 12-myristate 13-acetate (500 nM, 3 h) before replacing with fresh medium and leaving overnight.

Primary bone marrow-derived macrophages were prepared by isolating the bone marrow from the femurs and tibias of 3–6-month-old male and female wild-type C57BL/6 mice (Charles River), lysing red blood cells in ACK lysis buffer, and culturing the resulting bone marrow in 70% DMEM containing 10% FBS and 100 U ml$^{-1}$ penicillin and 100 µg ml$^{-1}$ streptomycin, supplemented with 30% L929-conditioned medium for 7 days. Before experiments, cells were seeded at a density of $1 \times 10^6$ cells ml$^{-1}$ and left overnight.

### Plasmids

Coding sequences were obtained from human *IL1A* (NCBI Gene ID: 3552), human *IL1B* (NCBI Gene ID: 3553) and *Orcinus orca IL1A* (NCBI reference sequence: XM_004276766.2). Pro-IL-1α, pro-α-mat-β, orca pro-(h)IL-1α and pro-IL-1β genes were synthesised and cloned to pcDNA3.1$^{(+)}$ vectors (Life Technologies). pro-α-mat-β chimeric expressed the Met[1] to Arg[112] of human IL-1α, followed by Tyr[113] to Ser[269] of human IL-1β. Orca pro-(h)IL-1α chimeric expressed the Met[1] to Ile[108] of Orca IL-1α, followed by Ile[109] to Ala[271] of human IL-1α. To generate IL-1α lentiviral expression vectors, Addgene plasmid #50919 was digested with BstBI and BstXI and a synthesised fragment comprising a multiple cloning site (MCS) and bGH polyA was inserted via restriction cloning, to create pLV-Ef1a-MCS. This plasmid was digested with KpnI and NotI and used as the backbone for the following constructs. Human pro-IL-1α (NLS coding sequence KKRR; aagaagagacgg) was amplified from pcDNA3.1-pro-IL-1α, using F1F and F2R. The NLS mutant (NLS coding sequence KKRW; aagaagagaTgg) was generated via a two-fragment HiFi assembly using primers to introduce the mutation. Fragment 1, F1F and aF1R, Fragment 2, aF2F and F2R. The second NLS mutant (NLS coding sequence KNRW; aagaaCagaTgg) was again generated by two fragment HiFi assembly using primers F1F and bF1R for fragment 1, and bF2F and F2R for fragment 2. To generate a pro-IL-1α-

TurboID expression vector, the pro-IL-1α sequence was amplified from pcDNA3.1-pro-IL-1α using pF1F and pF1R, and TurboID sequence amplified from Addgene plasmid #107173 using primers tF2F and tF2R. The vector was assembled by HiFi assembly (NEB) into pcDNA3.1/KpnI/NotI. To generate the TID expression vector, the TurboID sequence was amplified from Addgene plasmid #107173 using primers tF1F and tF1R, followed by HiFi assembly into pLV-Ef1a-MCS/KpnI/NheI. All primer sequences are available in Supplementary Data 3. Plasmids are available from the authors upon reasonable request. Plasmid sequences are available in Supplementary Data 3.

### Transient transfections
For experiments using the chimeric IL-1 constructs, HeLa cells were transfected for 18–24 h with 250 ng of human pro-IL-1α, human pro-IL-1β, human pro-α-mat-β, or Orca pro-IL-1α using Lipofectamine 3000, according to manufacturer's instructions. For experiments using the single or double amino acid mutant pro-IL-1α constructs, HeLa cells were transfected for 24 h with 100 or 250 ng of full-length human WT pro-IL-1α, or pro-IL-1α containing the KKRW or KNRW mutations using FuGENE HD transfection reagent at a ratio of 3:1 or Lipofectamine 3000 according to manufacturer's instructions. For TurboID experiments, HeLa cells were transfected for 18 h with either pro-IL-1α (500 ng), pro-IL-1α-TurboID (500 ng) or TurboID alone (125 ng) per $5 \times 10^4$ cells seeded using Lipofectamine 3000. For mass spectrometry analysis, after transfection of cells with pro-IL-1α-TurboID or TurboID alone, cell media was changed to complete DMEM and left to incubate for a further 24 h prior to stimulation to obtain an optimal cell number for stimulation and cell scraping.

### Immunocytochemistry
Cells were washed once in PBS then fixed in 4% paraformaldehyde for 15 min before being permeabilized with PBS, 0.1% Triton-X-100 (PBST) for 5 min at room temperature. Cells were incubated in blocking solution (5% BSA, PBST) for 1 h at room temperature, before incubation with goat anti-human IL-1α (2 μg ml⁻¹), rabbit anti-human BirA (1:500 v/v), streptavidin-HRP (1 μg ml⁻¹), rabbit anti-human EP300 (4.6 μg ml⁻¹), rabbit anti-human ZZZ3 (9.3 μg ml⁻¹), rabbit anti-human ANKRD17 (2 μg ml⁻¹) or goat anti-mouse IL-1α (2 μg ml⁻¹) primary antibodies in 1% BSA PBST at 4 °C overnight. Cells were then incubated with Alexa Fluor™ 488 donkey anti-goat IgG (4 μg ml⁻¹), Alexa Fluor™ 488 donkey anti-rabbit IgG (4 μg ml⁻¹), Alexa Fluor™ 594 donkey anti-goat IgG (4 μg ml⁻¹), Alexa Fluor 647™ donkey anti-rabbit IgG (4 μg ml⁻¹) or Streptavidin Alexa Fluor™ 594 conjugate (2 μg ml⁻¹) secondary antibodies for 1 h at room temperature. Nuclei were stained with DAPI (0.5 μg ml⁻¹). Coverslips were mounted on slides using ProLong Gold antifade mounting reagent and left to dry overnight.

### Microscopy and nuclear localisation analysis
Widefield microscopy images were acquired for experiments using the chimeric IL-1 constructs using a Zeiss Axioimager.D2 upright microscope using a ×20/0.5 EC Plan-neofluar objective and captured using a Coolsnap HQ2 camera (Photometrics) through Micromanager software v1.4.23. Specific band pass filter sets for DAPI, FITC and Texas red were used to prevent bleed through from one channel to the next. Images were acquired from 3–5 fields of view from each independent experiment, and nuclear localisation was quantified using the ImageJ 'Intensity Ratio Nuclei Cytoplam' tool and expressed as percentage of total IL-1 fluorescence intensity that co-localised with DAPI signal.

Confocal microscopy images were acquired using a ×63/1.40 HCS PL Apo objective on a Leica TCS SP8 AOBS inverted or upright confocal microscope with LAS X software (v3.5.1.18803). To prevent interference between channels, lasers were excited sequentially for each channel. The blue diode with 405 nm and the white light laser with 488 nm, 594 nm, and 647 nm laser lines were used, with hybrid and photon-multiplying tube detectors with detection mirror settings set

appropriately. Z-stacks were acquired with 0.3 μm steps between Z sections. Images were acquired from 3–5 fields of view from each independent experiment, and pro-IL-1α nuclear localisation was quantified manually using maximum projections and expressed as the percentage of total pro-IL-1α fluorescence intensity that co-localised with DAPI signal. Pearson's correlation coefficient was determined using the Coloc 2 plugin (ImageJ), using the IL-1α signal as a region of interest on maximum intensity projections.

### IL-1α release experiments
Following HeLa cell transfection as described above, cells were treated with ionomycin (10 μM) for 1 or 4 h in serum-free DMEM. IL-1α release into the supernatant was measured by enzyme-linked immunosorbent assay (ELISA) according to manufacturer's instructions. Cell death was determined by measuring lactate dehydrogenase (LDH) release into the supernatant using CytoTox 96 nonradioactive cytotoxicity assay according to manufacturer's instructions.

### Biotinylation experiments
Prior to mass spectrometry analysis, HeLa cells transfected with pro-IL-1α-TurboID or TurboID alone were treated with biotin (500 μM) for 30 min in serum-free DMEM. Cells were then washed four times in cold PBS (containing CaCl₂ and MgCl₂), scraped and pelleted at 1000 × g for 5 min at 4 °C. Prior to streptavidin pull-down for the proximity labelling validation experiments, HeLa cells transfected with pro-IL-1α-TurboID or TurboID alone were treated with biotin (500 μM) for 60 min in optimem. Cells were then washed 3 times in cold PBS (containing CaCl₂ and MgCl₂) before lysis in RIPA buffer containing protease inhibitor cocktail. Cell lysates were clarified by centrifugation at 15,000 × g at 4 °C, the input was collected, and the remaining lysate was incubated with equilibrated MagReSyn streptavidin beads overnight at 4 °C. The following day, the beads were pelleted using a magnetic rack and the flow-through was collected. Beads were sequentially washed in RIPA buffer, KCl (1 M), Na₂CO₃ (0.1 M), urea (2 M) and again in RIPA buffer, prior to elution in 2× Laemmli buffer containing DTT (30 mM) and biotin (2 mM) at 70 °C.

### MaxQuant analysis, filtering, and transformation
Streptavidin pull-down and mass spectrometry analysis was commercially outsourced and carried out by Sanford Burnham Prebys Proteomics Core, La Jolla. Mass spectrometry was carried out on dried and isolated biotin-enriched peptides following biotinylation in pro-IL-1α-TurboID or TurboID groups (n = 4). Cells were lysed in 8 M urea, 50 mM ammonium bicarbonate and benzonase, and the lysate was centrifuged at 14,000 × g for 15 min to remove cellular debris. Supernatant protein concentration was determined using a bicinchoninic acid protein assay (ThermoFisher). Disulphide bridges were reduced with 5 mM tris(2-carboxyethyl)phosphine at 30 °C for 60 min, and cysteines were subsequently alkylated with 15 mM iodoacetamide in the dark at room temperature for 30 min. Affinity purification was carried out in a Bravo AssayMap platform (Agilent) using AssayMap streptavidin cartridges (Agilent). Briefly, cartridges were first primed with 50 mM ammonium bicarbonate, and then proteins were slowly loaded onto the streptavidin cartridge. Background contamination was removed with 8 M urea, 50 mM ammonium bicarbonate. Finally, cartridges were washed with Rapid digestion buffer (Promega) and proteins were subjected to on-cartridge digestion with mass spec grade Trypsin/Lys-C Rapid digestion enzyme (Promega) at 70 °C for 1 h. Digested peptides were then desalted in the Bravo platform using AssayMap C18 cartridges, and dried down in a SpeedVac concentrator.

Prior to LC-MS/MS analysis, dried biotin-enriched peptides were reconstituted with 2% ACN, 0.1% FA and concentration was determined using a NanoDrop™ spectrophometer (ThermoFisher). Samples were then analysed by LC-MS/MS using a Proxeon EASY-nanoLC system (ThermoFisher) coupled to a Orbitrap Fusion Lumos Tribid mass

spectrometer (ThermoFisher). Peptides were separated using an analytical C18 Aurora column (75 μm × 250 mm, 1.6 μm particles; IonOpticks) at a flow rate of 300 nL/min (60 °C) using a 75 min gradient: 2% to 6% B in 1 min, 6% to 23% B in 45 min, 23% to 34% B in 28 min, and 34% to 48% B in 1 min (A = FA 0.1%; B = 80% ACN: 0.1% FA). The mass spectrometer was operated in positive data-dependent acquisition mode. MS1 spectra were measured in the Orbitrap in a mass-to-charge (m/z) of 375–1500 with a resolution of 60,000. Automatic gain control target was set to $4 \times 10^5$ with a maximum injection time of 50 ms. The instrument was set to run in top speed mode with 1 second cycles for the survey and the MS/MS scans. After a survey scan, the most abundant precursors (with charge state between +2 and +7) were isolated in the quadrupole with an isolation window of 0.7 m/z and fragmented with HCD at 30% normalised collision energy. Fragmented precursors were detected in the ion trap as rapid scan mode with automatic gain control target set to $1 \times 10^4$ and a maximum injection time set at 35 ms. The dynamic exclusion was set to 20 s with a 10 ppm mass tolerance around the precursor.

Mass spectrometry data were analysed with MaxQuant software (v1.6.11.0). MS/MS spectra were searched against the *Homo sapiens* Uniprot protein sequence database (downloaded in April 2022) and GPM cRAP sequences (commonly known protein contaminants). Precursor mass tolerance was set to 20 ppm and 4.5 ppm for the first search where initial mass recalibration was completed and for the main search, respectively. Product ions were searched with a mass tolerance 0.5 Da. The maximum precursor ion charge state used for searching was 7. Carbamidomethylation of cysteine was searched as a fixed modification, while oxidation of methionine and acetylation of protein N-terminal were searched as variable modifications. Enzyme was set to trypsin in a specific mode and a maximum of two missed cleavages was allowed for searching. The target-decoy-based false discovery rate (FDR) filter for spectrum and protein identification was set to 1%. Quantification was provided as LFQ (label-free quantification) intensities for both peptides and protein groups. LFQ intensities for protein groups were used as the unit for investigation from this point forward. The Perseus computational platform[37] (https://maxquant.net/perseus/) was used to prepare protein group LFQ intensities for pro-IL-1α-TurboID and TurboID. Protein groups were loaded into Perseus and LFQ intensities were identified as the columns of interest in this analysis. The dataset was then filtered to remove potential contaminants, reverse hits and proteins only identified by site. Data were transformed using a $\log_2(x)$ transformation and normal distribution was checked by plotting histograms of LFQ intensity for all repeats within each experimental group (pro-IL-1α-TurboID or TurboID alone). Samples were then grouped according to biological replicates and rows were filtered so that at least one experimental group (pro-IL-1α-TurboID or TurboID alone) contained a valid value for all four replicates. Imputation was carried out to allow any further missing values to be replaced with values predicted from a normal distribution. PCA was performed on each biological replicate within pro-IL-1α-TurboID and TurboID alone groups. Enrichment was determined by two-sample two-tailed $t$ test of $\log_2$-transformed LFQ intensity values (significance determined as s0 = 2; FDR = 0.01) using Perseus computational platform.

### Ingenuity pathway analysis
Ingenuity pathway analysis (IPA, Qiagen) was used to analyse pathway enrichment in pro-IL-1α-TurboID significantly enriched proteins. Subcellular location analysis was achieved by IPA using gene ontology (GO) database of subcellular location terms. For canonical pathway and biological function analysis, the p-value of overlap was calculated according to the overlap between pro-IL-1α-TurboID significantly enriched proteins and proteins present with each pathway/function. P-values were determined by right-tailed Fisher's exact test using IPA software, and p < 0.01 was used as a significance threshold. "Number of

proteins" denotes the number of pro-IL-1α-TurboID significantly enriched proteins that are present within a canonical pathway or biological function as determined by IPA. Biological function pathways shown are the top 10 significant pathways following a significance threshold of p < 0.01.

### STRING analysis
Gene names for proteins significantly enriched in pro-IL-1α-TurboID were uploaded into STRING. Network shows "physical subnetwork" meaning all physical interactions previously reported in the literature and functional annotations were exported containing all gene ontology terms associated with the pro-IL-1α-TurboID proximity-based interactome. HAT proteins were determined by filtering gene ontology terms for "containing: histone, and containing: acetyltransferase".

### Western blotting
Western blot analysis was performed on HeLa cell lysates and supernatants for IL-1α, BirA, streptavidin, EP300, ZZZ3 and NCAPH. Supernatant was collected and cells were lysed in lysis buffer (50 mM Tris-HCl, 150 mM NaCl; Triton-X-100 1% v/v, pH 7.3) containing protease inhibitor cocktail, or according to streptavidin pull-down protocol described earlier. Equal volumes of lysates or supernatants were run on SDS-polyacrylamide gels or NuPAGE tris-acetate gels, and transferred at 25 V onto PVDF membranes using a Trans-Blot® Turbo Transfer™ System (Bio-Rad). Membranes were blocked in 5% BSA (w/v) in PBS, 0.1% Tween (v/v) for 1 h at room temperature. Membranes were incubated at 4 °C overnight with goat anti-human IL-1α (200 ng ml⁻¹), rabbit anti-human BirA (1:1000 v/v), streptavidin-HRP (500 ng ml⁻¹), rabbit anti-human EP300 (460 ng ml⁻¹), rabbit anti-human ZZZ3 (930 ng ml⁻¹) or rabbit anti-human NCAPH (600 ng ml⁻¹) in 5% BSA (w/v) in PBS, 0.1% Tween. Membranes were washed in PBS Tween and incubated in rabbit anti-goat IgG (500 ng ml⁻¹) or goat anti-rabbit IgG (250 ng ml⁻¹) in 5% BSA (w/v) in PBS, 0.1% Tween for 1 h at room temperature. Proteins were visualised with Cytiva Amersham ECL Prime Western Blotting Detection Reagent and G:BOX (Syngene) and Genesys software. Membranes were probed for β-actin as a loading control. Densitometry was performed using ImageJ. The total IL-1α signal was calculated as the combined densitometric signal of pro-IL-1α and mature IL-1α bands in both the supernatant and lysate.

### Protein alignment and tree analysis
IL-1α amino acid sequences were retrieved from the NCBI orthologues database (Ref_Seq) (accessed 05/08/2021). In total, 226 amino acid sequences from 157 species were obtained, including all available isoforms from each species. Sequences were aligned using MUSCLE on MEGAX using default settings[38] and the NLS sequences (⁷⁸GKVLKKRRLSLSQ⁹⁰ in human) were analysed. The full protein sequence alignment is available, including all species and accession numbers (Supplementary Data 1).

Model evolutionary trees were separately manually generated using species divergence times retrieved from TimeTree[17]. For phylogenetic tree construction using IL-1α protein sequences, the evolutionary history was inferred by using the Maximum Likelihood method and JTT matrix-based model[39]. The tree with the highest log likelihood (−7883.54) is shown. The percentage of trees in which the associated taxa clustered together is shown next to the branches, from 1000 bootstrap replications[40]. Initial tree(s) for the heuristic search were obtained automatically by applying Neighbour-Join and BioNJ algorithms to a matrix of pairwise distances estimated using the JTT model, and then selecting the topology with superior log likelihood value. A discrete Gamma distribution was used to model evolutionary rate differences among sites (5 categories (+G, parameter = 1.8987)). The tree is drawn to scale, with branch lengths measured in the number of substitutions per site. This analysis involved 30 amino acid sequences. All positions with less than 95% site coverage were eliminated, i.e.,

fewer than 5% alignment gaps, missing data, and ambiguous bases were allowed at any position (partial deletion option). There were a total of 248 positions in the final dataset. Evolutionary analyses were conducted in MEGAX[41].

HAT-binding domain conservation was assessed by comparing the aligned IL-1α pro-domain sequences to the modal amino acid sequence determined from the full amino acid sequence alignment (see Supplementary Data 1). The HAT-binding domains previously identified in pro-IL-1α (amino acids 7-19 and 98-108 in human pro-IL-1α[10]) were compared between species with a functional and non-functional NLS.

### NLS score
NLS score was calculated using NLS mapper[42] using the full IL-1α amino acid sequence, or part of the exon 3 amino acid sequence obtained from the SRA analysis below, selecting the highest monopartite NLS score containing the KKRR motif region.

### SRA analysis
We obtained DNA and RNA sequencing read data for the mutant NLS species, as well as closely related species, from the NCBI sequence read archive database (https://www.ncbi.nlm.nih.gov/sra). All sequence accession numbers are available in Supplementary Data 2. We then used sequence read archive nucleotide BLAST (default parameters, optimised for megablast or blastn) to align these sequence reads against an IL-1α exon 3 nucleotide query sequence (obtained from NCBI genome viewer) of the respective closely related NLS mutant species, to identify reads that closely matched the query sequence. Where possible, up to three sequence reads were obtained from different animals, from different studies. We then converted up to the top 100 sequence reads that had an alignment score >80 to a multiple sequence alignment using the NCBI multiple sequence alignment viewer. For each nucleotide position coding for the KKRR motif in the IL-1α NLS (aagaagagacgg in human), the percentage of bases in the sequence reads that matched the human sequence base was calculated. When necessary, nucleotide sequences were translated using https://web.expasy.org/translate/.

### Statistics and reproducibility
Nuclear localisation quantification and Pearson's correlation coefficient analysis from immunofluorescence images are presented as median ± interquartile range, with each data point representing a field of view. IL-1α release data are presented as mean ± SEM. Data were assessed for normal distribution using Shapiro–Wilk normality test. Parametric data were analysed using one-way ANOVA with Dunnett's post-hoc test. Non-parametric data were analysed using unpaired two-tailed Mann–Whitney test or Kruskal–Wallis test with Dunn's post-hoc test. Data normalised as a percentage were analysed using Wilcoxon signed-rank test versus a value of 100% or multiple Mann-Whitney tests versus a value of 100% followed by Holm-Sidak correction. TurboID experiments were analysed as stated in the methods and figure legends. Western blots are representative of 4–6 independent experiments. Statistical analysis was performed using GraphPad Prism (v9). No statistical method was used to predetermine sample size. No data were excluded from the analyses. The experiments were not randomised. The investigators were blinded to treatments or analysis where appropriate.

### Reporting summary
Further information on research design is available in the Nature Portfolio Reporting Summary linked to this article.

## Data availability
The data that support the findings of this study are available within the article, Supplementary Figs. and supplementary data files. Uncropped western blots are provided in the source data file and in Supplementary Fig. 18. Proteomics mass spectrometry raw data are available at the ProteomeXchange Consortium via the PRIDE partner repository under the accession code PXD053438, and processed log₂-transformed LFQ values are available in the source data file. All IL-1α amino acid sequence accession codes are listed in Supplementary Data 1. All sequence read archive accession codes are listed in Supplementary Data 2. Source data are provided in this paper.

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

## Acknowledgements
D.B., C.H., P.I.S., G.C. and M.L. were funded by Medical Research Council (MRC) grant MR/T016515/1. R.W. was funded by BHF Accelerator Award AA/18/4/34221. V.S.T. was funded by CONICYT (Becas Chile 721704488). J.P.G. was funded by MRC grant MR/W028867/1. G.L.-C. was funded by MRC grant MR/T016043/1. The Bioimaging Facility microscopes used in this study were purchased with grants from BBSRC, Wellcome and the University of Manchester Strategic Fund. Protein mass spectrometry was performed by Sanford Burnham Prebys Proteomics Core.

## Author contributions
Conceptualisation: D.B., G.L.-C. M.L. and C.H. Methodology: H.B. and A.A. Investigation: R.W., V.S.T., P.I.S., G.C. and C.H. Visualisation: R.W. and C.H. Funding acquisition: D.B. and G.L.-C. Supervision: D.B., G.L.-C., J.P.G. and C.H. Writing - original draft: R.W., C.H., and D.B. Writing–review and editing: R.W., C.H., D.B., J.P.G., V.S.T., P.I.S., G.L.-C., J.R.-A., H.B. and A.A.

## Competing interests
The authors declare no competing interests.

## Additional information

[1]Division of Neuroscience, School of Biological Sciences, Faculty of Biology, Medicine and Health, University of Manchester, Manchester Academic Health Science Centre, Manchester M13 9PT, UK. [2]Geoffrey Jefferson Brain Research Centre, The Manchester Academic Health Science Centre, Northern Care Alliance NHS Group, University of Manchester, Manchester M13 9PT, UK. [3]The Lydia Becker Institute of Immunology and Inflammation, University of Manchester, Manchester M13 9PT, UK. [4]Genome Editing Unit, Faculty of Biology, Medicine and Health, University of Manchester, Manchester M13 9PT, UK. [5]Tasmanian School of Medicine, College of Health and Medicine, University of Tasmania, Hobart, TAS 7000, Australia. [6]Division of Molecular and Cellular Function, School of Biological Sciences, Faculty of Biology, Medicine and Health, University of Manchester, Manchester Academic Health Science Centre, Manchester M13 9PT, UK. [7]Division of Infection, Immunity and Respiratory Medicine, School of Biological Sciences, Faculty of Biology, Medicine and Health, University of Manchester, Manchester Academic Health Science Centre, Manchester M13 9PT, UK. ✉e-mail: david.brough@manchester.ac.uk; christopher.hoyle@manchester.ac.uk

