## [Peer Review File · Nature Communications]

Proximity labelling of pro-interleukin-1 α reveals evolutionary conserved nuclear interactionsREVIEWER COMMENTS

Reviewer #1 (Remarks to the Author):

In this interesting paper the authors explore the mammalian diversity of interleukin 1 alpha focussing on the pro-domain and two key regions: the nuclear localisation sequence (NLS) and the histone acetyl transferase (HAT) binding domain. The biology of both IL1a and particularly its pro-domain are very much under explored. The interesting differences identified between mammals by the detailed evolutionary analysis in this paper proposes that the HAT domain and nuclear localisation domain evolved together, but some species lost the NLS whilst the HAT domain was maintained. The authors propose that loss of the NLS may confer an evolutionary advantage to animals that have lost it.

Questions for the authors to address

1. The hypotheses in this paper are very interesting, but currently remain theories. The paper would be strengthened by experimentally investigating some of the biology in the HeLa or a similar reconstitution cell model. Are there changes in HAT modification/ function in cells without the pro-IL1a or in cells where the HAT domains of pro-IL1a are mutated?
2. Is there a difference in nuclear HAT modification in cells where the NLS is mutated out?
3. Does loss of an NLS in human pro-IL1a alter its secretion compared to WT IL1a?

Reviewer #2 (Remarks to the Author):

Authors Rose Wellens et al. provide in their newly submitted manuscript interesting insights in IL-1 α function and evolution. Although presenting novel and very interesting results this manuscript needs a major revision before it can be considered further for publication in Nature Communications.

Following points need to be considered by authors in preparing a revised version:

Abstract lines 31-32 improvement is needed. "...multiple mammalian species" need more detailed description already in the abstract as well as "in species that had lost...localisation"

Introduction Lines 40-41: This needs a more comprehensive introductory description for non-specialist readers of all available molecules used for signaling purposes between immune cells (other chemokines, growth factors etc.)

Line 44 - in Reference #3 it is described as a superfamily, here it is considered only as a family? Do authors see no difference between family and superfamily? Although maybe controversial in the literature authors need to be consistent in their formulations...

Line 46 - "IL-1 β is an ancient protein" from the evolutionary point of view this cannot be stated in such a manner. There are many contemporary IL-1 β proteins that differ from the real ancient version that occurred earlier in the long term history. Maybe more description of the "ancient protein" of this (super)family is needed here.

Lines 54-56 From this description it is not fully clear for the reader how many independent domains are there actually in the whole protein of IL-1 α ?

Line 72 "manipulated" is not the right word at this place. If we consider that random mutations and selective forces are the evolutionary pressures this can be stated in a different way.

Line 137 "Thyroid receptor and retinoic acid receptor signalling pathways" as these pathways are very important it needs more explanation about some logical connection? Maybe some appropriate citation at this place?

Line 186 "toothed whale clade" although authors refer to their previous publication more detailed taxonomical description of this clade in terms of order and suborder is required at this place mainly for non-experts.

Lines 188-190 IMPORTANT The whole protein sequence alignment needs to be deposited in supplementary material for the overview. If 226 unique sequences (accession numbers?) were used from 157 different mammalian species then some paralogs were also included? This needs not to pose a significant problem but it shall be made available.

Lines 194-195 Taxonomical latin names need to be mentioned here also in connection with particular sequencing projects - also in connection with line 212.

Line 216 - is there some appropriate citation about the dating?

Line 224 -"isolated, separate evolutionary events" in this connection the mutational rate in other parts of IL genes would be interesting to compare.

Figure 4 IMPORTANT: No bootstrap values of the trees are presented here. From the phylogenetic point of view this is important to statistically support the presented claims. It is also not clear whether the "expanded model trees" were produced separately?

Figure 6 and Line 342 what are "modal amino acid sequences" in this context? Needs an explanation.

Lines 352-356 such considerations need to be supported by the e.g. bootstrap values of reconstructed trees as addressed already above in connection with Figure 4.

Discussion lines 405-406 although authors mutated the human gene appropriately it emerges the

question how high is the chance to investigate these IL functions in other native non-human mammalian cell lines?

Lines 587-588 again - the full protein sequence alignment needs to be made available in the supplementary material. Why only default settings were used? Also details on the construction of the phylogenetic tree in Figure 4 need to be described here.

Reviewer #3 (Remarks to the Author):

The manuscript by Wellens et al. utilized proximity labeling and sequence alignment analysis to show the intracellular role of pro-IL-1 α and the mutation in the NLS in different mammalian species. The author first expressed pro-IL-1 α -TurboID in HeLa cells and confirmed that the TurboID did not affect the localization and trafficking function of pro-IL-1 α . Then, the author used proximity labeling to identify the interactome of pro-IL-1 α , which included several HATs. The author further aligned pro-IL-1 α sequences from different mammalian species and identified mutations in the NLS. With the motif sequence in hand, the authors demonstrated that KKRR motif mutations in the NLS cause reduced nuclear localization of pro-IL-1 α . Meanwhile, the author found at least one HAT binding domain was highly conserved in most mammalian species. Finally, the author analyzed the IL-1 α region in monotremes and claimed the evolution of the NLS and possibly the N-terminal HAT-binding domain occurred after the IL-1 gene duplication event in the proto-mammal but before mammalian divergence.

Overall, the manuscript is well organized and written and shows that the KKRR motif mutation in the NLS causes reduced nuclear localization of pro-IL-1 α . However, the author has not provided sufficient new insights into the intracellular role of pro-IL-1 α (e.g. the biological consequence of the interaction between pro-IL-1 α and HAT proteins). In addition, some conclusions based on the results of the analyses should be adequately addressed. Most importantly, some additional experiments in primary systems are needed to support the biological significance of the new discovery in the IL-1 α networks.

Below are my concerns:

Major points:

1. The author used proximity labeling to identify 10 HAT proteins that interact with pro-IL-1 α , and only EP300 was reported. To further demonstrate the interaction between the other 9 HAT proteins and pro-IL-1 α , I wonder if the author could select one of the nine HAT proteins as an example to confirm the interaction between the HAT protein and pro-IL-1 α using the biochemical approach.
2. The author claimed that while some species lost the NLS in their pro-IL-1 α , but the HAT binding was maintained. Dose the author perform proximity labeling on the mutant human pro-IL-1 α (e.g. KNRW variant) to identify the enriched proteins and compare the enriched proteins with those in pro-IL-1 α group? This may shed light on how the mutation in the NLS shapes the function of pro-IL-1 α .
3. In Fig. 1E, treatment of pro-IL-1 α - and pro-IL-1 α -TurboID-expressing HeLa cells with ionomycin

for different durations showed a gradual decrease in pro-IL-1 α /pro-IL-1 α -TurboID levels. However, why mature IL-1 α /IL-1 α -TurboID did not show a significant increase with increasing treatment time remains unclear.

4. In Fig. 2C, the authors identified the pro-IL-1 α interactome containing 56 proteins. I noticed that one protein stands out as the most significant (with the highest fold change). This protein is likely an important candidate to actually interact with pro-IL-1 α . The authors can annotate the names of some enriched important proteins in the volcano plot.

5. When investigating the effect of NLS mutations on the nuclear localization of pro-IL-1 α , the authors performed their validation in cell lines (HeLa) expressing different variants of pro-IL-1 α (Fig. 5). However, it is unclear whether there is any other evidence or literature reporting the nuclear localization of pro-IL-1 α in primary cells of the species mentioned in the manuscript.

6. In most mammalian species, one of the HAT binding domains (amino acid residues 7-19) is more conserved, while the other (amino acid residues 98-108) is less conserved (Fig. 6). What are the biological implications of this mutation? Does the HAT binding domain have a conserved sequence similar to the KKRR motif in the NLS of pro-IL-1 α , and does mutation of such a conserved sequence result in loss of HAT binding function?

Minor points:

1. There is no scale bar for the microscopy images in Figure 5.
2. The author need to highlight proteins discovered in Figure 2C.

Reviewer #4 (Remarks to the Author):

This study attempted to employ TurboID-based proximity labeling to analyze the proximal interactome of Interleukin (IL)-1 α in order to understand the evolution in IL-1 α 's nuclear localization signal. Although several proteins in proximity to IL-1 α were identified through proximity labeling, there is insufficient experimental evidence to confirm whether these proteins indeed bind to IL-1 α . Additionally, the logical transition between the first part of the study, which focused on proximity labeling analysis, and the second part, which examined the evolution of the nuclear localization signal, seems somewhat disjointed, lacking a strong overall coherence.

Reviewer #1 (Remarks to the Author):

In this interesting paper the authors explore the mammalian diversity of interleukin 1 alpha focussing on the pro-domain and two key regions: the nuclear localisation sequence (NLS) and the histone acetyl transferase (HAT) binding domain. The biology of both IL1a and particularly its pro-domain are very much under explored. The interesting differences identified between mammals by the detailed evolutionary analysis in this paper proposes that the HAT domain and nuclear localisation domain evolved together, but some species lost the NLS whilst the HAT domain was maintained. The authors propose that loss of the NLS may confer an evolutionary advantage to animals that have lost it.

Questions for the authors to address

1. The hypotheses in this paper are very interesting, but currently remain theories. The paper would be strengthened by experimentally investigating some of the biology in the HeLa or a similar reconstitution cell model. Are there changes in HAT modification/ function in cells without the pro-IL1a or in cells where the HAT domains of pro-IL1a are mutated?

Previous studies have suggested that pro-IL-1 α can directly interact with HATs such as EP300, and that this interaction can regulate transcriptional activation (PMIDs: 14612453, 22879895, 24024155). Deletion of the N- and C-terminal regions of the pro-IL-1 α pro-domain (which contain the HAT-binding domains) abolished the interaction with EP300, and this resulted in reduced transcriptional activity in a luciferase activity assay (PMID: 14612453). We have added to the text in the discussion: 'We identified nine histone-modifying enzymes as part of the pro-IL-1 α proximity-based interactome, in addition to the HAT protein, EP300, which is reported to functionally interact with pro-IL-1 α in the nucleus to influence EP300-mediated transcriptional activity^{10,12,14}. Deletion of the HAT-binding domain regions in pro-IL-1 α abolished the interaction between pro-IL-1 α and EP300 and reduced this transcriptional activity¹⁰.'

In order to experimentally address whether the presence of IL-1 α affected HAT activity, we used a fluorometric HAT activity assay. Here, we transfected HeLa cells with mock transfection, or with WT human pro-IL-1 α or the KNRW mutant human pro-IL-1 α . We included a transfection with pro-IL-1 β to control for any possible effect of the transfection. Following this transfection, we collected the cell lysates, isolated the cytoplasmic and nuclear fractions, and analysed HAT activity using a fluorometric assay, in which acetyl groups are transferred from acetyl-CoA to histone H3 peptide by any active HATs present in the cell lysate. We saw no clear effect of pro-IL-1 α transfection on HAT activity in this assay in either the nuclear or cytoplasmic fractions (Figure A below). The slight reduction in HAT activity observed in the pro-IL-1 α -transfected cells in Figure A-A was also observed when cells were transfected with pro-IL-1 β , suggesting this decrease was a consequence of the transfection, likely due to low levels of cell death. This difference was less pronounced when we normalised for the amount of protein loaded into the reaction, and in fact this normalisation revealed that HAT activity slightly increased as a result of the transfection (Figure A-B). We confirmed that the transfection of pro-IL-1 α and pro-IL-1 β was successful (Figure A-C).

Figure A. HAT activity was not significantly affected by the presence of pro-IL-1 α . HeLa cells were transfected with mock, WT pro-IL-1 α , KNRW pro-IL-1 α , or pro-IL-1 β for 18-24h ($n=4$). (A) Cytoplasmic and nuclear fractions were isolated (using ab113474), and HAT activity was measured (using ab204709) over 30 minutes. (B) HAT activity was normalised per minute per μg protein in the sample. (C) HeLa cells were analysed by fluorescence microscopy to confirm the presence of pro-IL-1 α and pro-IL-1 β . Representative maximum projection confocal immunofluorescence images are shown. Scale bars are 20 μm . Data are mean \pm SEM. Data were analysed using a Kruskal-Wallis test followed by Dunn's post-hoc test (versus WT pro-IL-1 α) (B).

This could mean that in our hands we did not have the correct conditions in which to see an effect. It is possible that pro-IL-1 α can only affect the activity of specific HATs, and thus the broad-spectrum HAT activity assay is not sensitive enough to detect subtle changes in HAT activity. Alternatively, any interaction between pro-IL-1 α and HATs may be transient, and therefore any effect of pro-IL-1 α on HAT activity may be lost during this assay, or the interaction may have been disturbed during preparation of the lysate samples. Given that we saw no effect on HAT activity by WT pro-IL-1 α , is it unsurprising that we also saw no effect of the mutant KNRW pro-IL-1 α construct.

In light of these data, we have adjusted the text throughout the manuscript to lessen the suggestion that IL-1 α directly modulates HAT activity.

2. Is there a difference in nuclear HAT modification in cells where the NLS is mutated out?

Please see our response to comment 1 above.

Furthermore, we only mutated the NLS in pro-IL-1 α , and did not mutate the HAT-binding domains. The mutation in the NLS reduces pro-IL-1 α nuclear localisation, but does not completely abolish it as pro-IL-1 α is a small enough protein to passively diffuse into the nucleus. Therefore, it is likely that pro-IL-1 α is still able to interact with HATs in the nucleus, even when the NLS is mutated, although this interaction is likely to be reduced. We have added new data in Supplementary Figure 14, in which we transfected HeLa cells with WT human pro-IL-1 α or the KKRW and KNRW pro-IL-1 α mutants, and then co-labelled for IL-1 α and EP300. EP300 is located in the nucleus of these cells and co-localises more strongly with WT pro-IL-1 α , but this co-localisation is reduced for the KKRW and KNRW pro-IL-1 α mutants. Thus, this suggests that there may be reduced interaction between pro-IL-1 α and nuclear HATs in the NLS mutant species.

We have added the following text to the manuscript: 'The loss of pro-IL-1 α nuclear localisation caused by the NLS mutations also resulted in reduced co-localisation with EP300 (Supplementary Figure 14).'

3. Does loss of an NLS in human pro-IL1a alter its secretion compared to WT IL1a?

To address this point we have now included data in Supplementary Figure 15A-H, where we transfected HeLa cells with WT human pro-IL-1 α , or with the orca pro-domain of IL-1 α (which lacks a functional NLS) fused to the mature domain of human IL-1 α , as in Figure 5B. We then treated those cells with ionomycin for one or four hours to induce IL-1 α processing and release. We observed an increase in IL-1 α cleavage into its mature form after one hour of ionomycin by western blotting, and both an increase in cleavage and release after four hours of ionomycin treatment. We also transfected HeLa cells with the human pro-IL-1 α plasmids that contained a KKRR (WT), KNRW, or KKRW NLS motif, and treated these cells with ionomycin for one hour (Supplementary Figure 15I-N). The mutations in the NLS again resulted in increased cleavage and release of IL-1 α . These data suggest that nuclear localisation of pro-IL-1 α may limit processing and release of IL-1 α .

We have added the following text to the manuscript: 'We also tested whether reduced pro-IL-1 α nuclear localisation would affect the cleavage and release of IL-1 α . HeLa cells were transfected with human pro-IL-1 α or the Orca IL-1 α pro-domain fused to the human IL-1 α mature domain and subsequently treated with ionomycin for one or four hours. Release of Orca pro-IL-1 α into the supernatant was not significantly increased after one hour of ionomycin treatment, but we observed increased cleavage of Orca pro-IL-1 α into its mature form, with no difference in ionomycin-induced cell death (Supplementary Figure 15A-E). Following four hours of ionomycin treatment, the release of Orca pro-IL-1 α into the supernatant was enhanced, again with no difference in ionomycin-induced cell death (Supplementary Figure 15F-H). We also transfected HeLa cells with the human pro-IL-1 α plasmids containing a WT (KKRR), KNRW, or KKRW NLS motif, and treated these cells with ionomycin for one hour. The NLS mutations resulted in enhanced IL-1 α release in response to ionomycin, with no difference in ionomycin-

induced cell death (Supplementary Figure 15I-K), as well as enhanced cleavage of pro-IL-1 α in the supernatant and lysate (Supplementary Figure 15L-N).'

Reviewer #2 (Remarks to the Author):

Authors Rose Wellens et al. provide in their newly submitted manuscript interesting insights in IL-1 α function and evolution. Although presenting novel and very interesting results this manuscript needs a major revision before it can be considered further for publication in Nature Communications.

Following points need to be considered by authors in preparing a revised version:

Abstract lines 31-32 improvement is needed. "...multiple mammalian species" need more detailed description already in the abstract as well as "in species that had lost...localisation"

We have altered the text in the abstract to include some of the species groups – ‘multiple mammalian species, including toothed whales, castorimorpha and marsupials. However, HAT-binding domains were conserved in those species that had lost pro-IL-1 α nuclear localisation.’ Due to the word count limitation on the abstract, we are unable to list each mammalian species individually.

Introduction Lines 40-41: This needs a more comprehensive introductory description for non-specialist readers of all available molecules used for signaling purposes between immune cells (other chemokines, growth factors etc.)

We have altered the text such that it now reads ‘Communication between immune cells is mainly driven by signalling molecules called cytokines, which are a large group of proteins that includes chemokines, and growth factors. Cytokines are usually secreted proteins that bind to receptors on target cells to elicit a signalling response.’

Line 44 - in Reference #3 it is described as a superfamily, here it is considered only as a family? Do authors see no difference between family and superfamily? Although maybe controversial in the literature authors need to be consistent in their formulations...

We have amended the text to now say ‘IL-1 superfamily’.

Line 46 - "IL-1 β is an ancient protein" from the evolutionary point of view this cannot be stated in such a manner. There are many contemporary IL-1 β proteins that differ from the real ancient version that occurred earlier in the long term history. Maybe more description of the "ancient protein" of this (super)family is needed here.

We have amended the text to remove ‘ancient’ and now say ‘IL-1 β is a protein that first arose more than 400 million years ago and is conserved throughout vertebrates...’

Lines 54-56 From this description it is not fully clear for the reader how many independent domains are there actually in the whole protein of IL-1 α ?

We have amended the text both here and at the start of the paragraph to try and clarify that full length IL-1 α consists of a pro-domain and a mature domain. Within the pro-domain, sub-domains have been identified that are the nuclear localisation sequence and the HAT-binding domains. There are no known sub-domains within the IL-1 α mature domain.

Line 72 "manipulated" is not the right word at this place. If we consider that random mutations and selective forces are the evolutionary pressures this can be stated in a different way.

We have amended the text to now say – ‘subcellular location has been subjected to evolutionary pressures...’

Line 137 "Thyroid receptor and retinoic acid receptor signalling pathways" as these pathways are very important it needs more explanation about some logical connection? Maybe some appropriate citation at this place?

The IPA analysis generated these canonical pathways based on pathways that the significant hits are known to be involved in, rather than specifically stating that these pathways are directly involved in pro-IL-1 α nuclear signalling. The explanation is likely to be that many of the proteins significantly enriched by pro-IL-1 α -TurboID were proteins that have an important nuclear role, including HAT proteins, which will be involved in many of these nuclear receptor signalling pathways.

To clarify this, we have re-ordered and amended some of the text, so that it now reads: 'IPA was also used to determine the canonical pathways and biological functions predicted to be associated with the pro-IL-1 α proximity-based interactome based on the representation of the 56 significant proteins within these processes (Figure 2F). Multiple nuclear receptor signalling pathways were significantly enriched, including the aryl hydrocarbon receptor, glucocorticoid receptor, vitamin D receptor, thyroid receptor and retinoic acid receptor signalling pathways. Further to this, gene expression was the most significant biological function associated with the pro-IL-1 α proximity-based interactome (Figure 2G). These data highlight an association between pro-IL-1 \$\alpha\$ and many nuclear proteins involved in the regulation of a broad array of gene expression pathways.'

Line 186 "toothed whale clade" although authors refer to their previous publication more detailed taxonomical description of this clade in terms of order and suborder is required at this place mainly for non-experts.

We have amended the text to say – 'we recently identified that within the order cetacea, the toothed whale clade (odontoceti) contained mutations in the pro-IL-1 α NLS 'KKRR' motif, ..., whereas the baleen whale clade (mysticeti) retained KKRR motif conservation.'

Lines 188-190 IMPORTANT The whole protein sequence alignment needs to be deposited in supplementary material for the overview. If 226 unique sequences (accession numbers?) were used from 157 different mammalian species then some paralogs were also included? This needs not to pose a significant problem but it shall be made available.

We have now included the full protein sequence alignment as Supplementary Data 1, including all the sequence accession numbers, and have indicated this in the results section ('for full alignment see Supplementary Data 1'). We have also updated the Figure 4 and Supplementary Figure 5 legends, as well as the protein alignment section in the materials and methods to state that 'the full protein sequence alignment is available, including all species and accession numbers (Supplementary Data 1)'.

Lines 194-195 Taxonomical latin names need to be mentioned here also in connection with particular sequencing projects - also in connection with line 212.

We have now added taxonomical latin names of the species in lines 194-195, and have added the latin family names of the families mentioned in line 212.

Line 216 - is there some appropriate citation about the dating?

We have now added citations for the dating, as well as more information about the estimated divergence time of castorimorpha from rodents.

Line 224 -"isolated, separate evolutionary events" in this connection the mutational rate in other parts of IL genes would be interesting to compare.

Whilst we think it would be very interesting to look at the rate of mutation in parts of other interleukin genes, we think that this is slightly beyond the scope of this study which is focusing on IL-1 α and the NLS.

Figure 4 IMPORTANT: No bootstrap values of the trees are presented here. From the phylogenetic point of view this is important to statistically support the presented claims. It is also not clear whether the "expanded model trees" were produced separately?

All of the trees presented in the manuscript are model trees only, manually created based on the estimated species divergence times from Kumar et al 2022 (PMID: 35932227). They are not true phylogenetic trees generated using the full IL-1 α amino acid sequences. We have updated the figure legends for all figures where model trees are present to clarify this point. We have also clarified that all model trees were produced separately, and that the KKRR motif sequences were retrieved either from the full protein alignment of IL-1 α in the 157 species (Figure 4B, Supplementary Data 1), or from the sequence read analysis using the SRA database (Figure 4C, D; Supplementary Figure 8A, 9A, 10A, 11A).

We have now separately generated a phylogenetic tree using the full IL-1 α amino acid sequences of the species presented in Figure 4A as well as the monotreme (platypus and echidna) IL-1 α sequences, including 1000 bootstrap replications. These data are now presented in Supplementary Figure 6 and the methods for the construction of this tree have been added to the methods.

Figure 6 and Line 342 what are "modal amino acid sequences" in this context? Needs an explanation.

By 'modal amino acid sequences', we are referring to the modal amino acid sequence determined from the full pro-IL-1 α amino acid alignment that we performed using 226 sequences from 157 mammalian species (now included as Supplementary Data 1, and the modal sequence is also shown in this document). We have now adapted the text in the Figure 6 legend, as well as at line 342, to clarify this.

Lines 352-356 such considerations need to be supported by the e.g. bootstrap values of reconstructed trees as addressed already above in connection with Figure 4.

We have added the monotreme IL-1 α sequences to the phylogenetic tree constructed in reply to your comment above about Fig 4A, and this is now presented in Supplementary Figure 6. The divergence of monotreme and marsupial IL-1 α from placental mammal IL-1 α was predicted with high confidence from 1000 bootstrap replications, and this was consistent with the predicted species divergence times from Kumar et al 2022 (PMID: 35932227).

Discussion lines 405-406 although authors mutated the human gene appropriately it emerges the question how high is the chance to investigate these IL functions in other native non-human mammalian cell lines?

Please see the response to reviewer 3 comment 5, which also addresses this comment.

Lines 587-588 again - the full protein sequence alignment needs to be made available in the supplementary material. Why only default settings were used? Also details on the construction of the phylogenetic tree in Figure 4 need to be described here.

We have now amended the text in the protein alignment section in the materials and methods, to state that 'the full protein sequence alignment is available, including all species and accession numbers (Supplementary Data 1)'. The default settings were used for the alignment using MUSCLE in MegaX as recommended by Hall et al 2013 (PMID: 23486614), and we have now added this reference to the manuscript. We have also added details about how the model phylogenetic trees were manually

generated based on estimated divergence times from Kumar et al 2022 (PMID: 35932227), as well as full details about how the new phylogenetic tree added as Supplementary Figure 6 was constructed using MEGAX.

Reviewer #3 (Remarks to the Author):

The manuscript by Wellens et al. utilized proximity labeling and sequence alignment analysis to show the intracellular role of pro-IL-1 α and the mutation in the NLS in different mammalian species. The author first expressed pro-IL-1 α -TurboID in HeLa cells and confirmed that the TurboID did not affect the localization and trafficking function of pro-IL-1 α . Then, the author used proximity labeling to identify the interactome of pro-IL-1 α , which included several HATs. The author further aligned pro-IL-1 α sequences from different mammalian species and identified mutations in the NLS. With the motif sequence in hand, the authors demonstrated that KKRR motif mutations in the NLS cause reduced nuclear localization of pro-IL-1 α . Meanwhile, the author found at least one HAT binding domain was highly conserved in most mammalian species. Finally, the author analyzed the IL-1 α region in monotremes and claimed the evolution of the NLS and possibly the N-terminal HAT-binding domain occurred after the IL-1 gene duplication event in the proto-mammal but before mammalian divergence.

Overall, the manuscript is well organized and written and shows that the KKRR motif mutation in the NLS causes reduced nuclear localization of pro-IL-1 α . However, the author has not provided sufficient new insights into the intracellular role of pro-IL-1 α (e.g. the biological consequence of the interaction between pro-IL-1 α and HAT proteins). In addition, some conclusions based on the results of the analyses should be adequately addressed. Most importantly, some additional experiments in primary systems are needed to support the biological significance of the new discovery in the IL-1 α networks. Below are my concerns:

Major points:

1. The author used proximity labeling to identify 10 HAT proteins that interact with pro-IL-1 α , and only EP300 was reported. To further demonstrate the interaction between the other 9 HAT proteins and pro-IL-1 α , I wonder if the author could select one of the nine HAT proteins as an example to confirm the interaction between the HAT protein and pro-IL-1 α using the biochemical approach.

We have now added an experiment (new Figure 3B) where we transfected HeLa cells with pro-IL-1 α -TID, or TID alone, and then treated the cells with biotin, followed by streptavidin pull down of the biotinylated proteins. We then performed western blotting on these samples to see if biotinylation of both EP300 and a second HAT protein identified in Figure 2-3, ZZZ3, was enriched by pro-IL-1 α -TurboID. ZZZ3 is a core subunit of the Ada-two-A-containing (ATAC) histone acetyl transferase complex (PMID: 30217978). We show that the biotinylation of both EP300 and ZZZ3 was enriched by pro-IL-1 α -TurboID compared to TurboID alone. These data suggest that pro-IL-1 α is either directly interacting with EP300 and ZZZ3, or is in extremely close proximity to these proteins, possibly in a complex with them. Furthermore, we have also blotted for NCAPH in this experiment, which was a protein identified in the mass spec experiment as being significantly enriched by TurboID (see Supplementary Figure 3C). Hence we believe this is a good control to show that not all proteins are enriched by pro-IL-1 α -TID.

We have also added data as Figure 3C-D, in which we have transfected HeLa cells with WT pro-IL-1 α , and performed immunofluorescence labelling for pro-IL-1 α and EP300 or ZZZ3. We show that pro-IL-1 α , EP300 and ZZZ3 all exhibit nuclear localisation, indicating that these proteins reside in the same subcellular compartment. As a control, in Figure 3E, we also stained for ANKRD17 in this experiment, which was a protein significantly enriched by TurboID (see Supplementary Figure 3C). ANKRD17 exhibited nuclear localisation as well as nuclear membrane localisation, suggesting that the biotinylation of EP300 / ZZZ3 by pro-IL-1 α -TID was specific, and not simply a result of both proteins being in the nucleus. We have also added a similar experiment in human THP-1 macrophages (Supplementary Figure 4), which were unprimed or primed with lipopolysaccharide to induce pro-IL-1 α expression. Similar to the HeLa experiment, pro-IL-1 α , EP300 and ZZZ3 were all present in the nucleus.

We have added the following text: 'In order to validate some of the significantly enriched protein hits from the proximity labelling experiment, we repeated the experiment by transfecting HeLa cells with pro-IL-1 α -TurboID and TurboID alone, followed by treatment with biotin. We then performed a streptavidin pull-down of the biotinylated proteins, before analysing the streptavidin-enriched eluates by western blotting (Figure 3B). Both pro-IL-1 α -TurboID and TurboID strongly biotinylated themselves, and therefore were strongly enriched in their respective eluates. The HAT protein EP300 was enriched in the eluate of cells transfected with pro-IL-1 α -TurboID compared to TurboID alone, as was ZZZ3, another HAT protein identified in the mass spectrometry analysis (Figure 3B) ¹⁶. Furthermore, we confirmed that NCAPH, a protein that we had identified in the mass spectrometry analysis as being significantly enriched by TurboID (Supplementary Figure 3C), was enriched in the eluate of cells transfected with TurboID alone (Figure 3B). Finally, immunofluorescence labelling revealed that both EP300 and ZZZ3 co-localised with pro-IL-1 α in the nucleus of HeLa cells (Figure 3C, D) and in human THP-1 macrophages (Supplementary Figure 4), indicating that these proteins reside in the same subcellular compartment, with EP300 exhibiting a homogeneous distribution throughout the nucleus, whereas ZZZ3 formed puncta. ANKRD17, a protein that was significantly enriched by TurboID alone (Supplementary Figure 3C), also exhibited nucleoplasmic and nuclear membrane localisation, and co-localised with pro-IL-1 α in the nucleus of HeLa cells (Figure 3E). Thus, nuclear localisation alone was not sufficient for significantly enriched biotinylation by pro-IL-1 α -TID.'

2. The author claimed that while some species lost the NLS in their pro-IL-1 α , but the HAT binding was maintained. Dose the author perform proximity labeling on the mutant human pro-IL-1 α (e.g. KNRW variant) to identify the enriched proteins and compare the enriched proteins with those in pro-IL-1 α group? This may shed light on how the mutation in the NLS shapes the function of pro-IL-1 α .

We have added new data in Supplementary Figure 14, in which we transfected HeLa cells with WT human pro-IL-1 α or the KKRW and KNRW pro-IL-1 α mutants, and then co-stained for IL-1 α and EP300. EP300 is located in the nucleus of these cells and co-localises more strongly with WT pro-IL-1 α , but this co-localisation is reduced for the KKRW and KNRW pro-IL-1 α mutants. Thus, these data suggest that there may be reduced interaction between pro-IL-1 α and nuclear HATs in the NLS mutant species.

3. In Fig. 1E, treatment of pro-IL-1 α - and pro-IL-1 α -TurboID-expressing HeLa cells with ionomycin for different durations showed a gradual decrease in pro-IL-1 α /pro-IL-1 α -TurboID levels. However, why mature IL-1 α /IL-1 α -TurboID did not show a significant increase with increasing treatment time remains unclear.

The blot presented in Figure 1E in the original submission was of cell lysates only. Ionomycin induces cell death, upon which IL-1 α gets released into the supernatant. To clarify this, we have now blotted supernatants of pro-IL-1 α - and pro-IL-1 α -TurboID-expressing HeLa cells after a timecourse of ionomycin treatment (see new Figure 1E in manuscript). We observed time-dependent release of IL-1 α (both the pro and mature forms) into the supernatant for both IL-1 α constructs, therefore explaining why mature IL-1 α and mature IL-1 α -TID did not continue to accumulate in the cell lysate over time. To ensure that the blots presented in Fig 1E are from the same experiment, we have changed the cell lysate blot accordingly.

4. In Fig. 2C, the authors identified the pro-IL-1 α interactome containing 56 proteins. I noticed that one protein stands out as the most significant (with the highest fold change). This protein is likely an important candidate to actually interact with pro-IL-1 α . The authors can annotate the names of some enriched important proteins in the volcano plot.

The protein that the reviewer has identified was in fact pro-IL-1 α itself, which we have now annotated on the volcano plot. We believe that pro-IL-1 α being the top hit actually validates the experiment, as the TurboID tag on pro-IL-1 α will strongly biotinylate pro-IL-1 α itself. We have now also annotated several

other top hits in the volcano plot, and all of the enriched proteins are listed in Supplementary Table 1 along with their fold change and p-value. Furthermore, we have also annotated the volcano plot in Supplementary Figure 3C with some of the proteins significantly enriched by TurboID alone. We have highlighted NCAPH in this figure, which is the nuclear protein enriched by TurboID that we blotted for in Figure 3B, as well as ANKRD17 from Figure 3E.

5. When investigating the effect of NLS mutations on the nuclear localization of pro-IL-1 α , the authors performed their validation in cell lines (HeLa) expressing different variants of pro-IL-1 α (Fig. 5). However, it is unclear whether there is any other evidence or literature reporting the nuclear localization of pro-IL-1 α in primary cells of the species mentioned in the manuscript.

It has been previously established in the literature that pro-IL-1 α exhibits localisation to the nucleus in human and mouse primary cells and cell lines (PMIDs: 8408068, 18939951, 23684408, 22206506). Thus, mouse cells can also be used to investigate pro-IL-1 α nuclear localisation. We have now added data demonstrating the nuclear localisation of IL-1 α in mouse primary bone marrow-derived macrophages upon priming with lipopolysaccharide (LPS), which induces pro-IL-1 α expression (Supplementary Figure 1). Furthermore, the immunofluorescence labelling added in Supplementary Figure 4 in human THP-1 macrophages also shows nuclear localisation of pro-IL-1 α in response to LPS priming. These data demonstrate IL-1 α nuclear localisation not just in mouse and human cells, but specifically in immune cells from these species, which is particularly relevant for IL-1 α signalling.

However, as far as we are aware, there is no current evidence regarding pro-IL-1 α nuclear localisation in primary cells or cell lines of the pro-IL-1 α NLS mutant species identified in this study (such as the castorimorpha, toothed whales, or marsupials), presumably in part due to lack of access to these cells, and a lack of tools such as antibodies to properly investigate these cells. Interestingly, it appears various cells such as epidermal and kidney cell lines have been derived from dolphins (e.g. PMIDs: 16281302, 37524977), along with other cell lines from cetaceans and marsupials (e.g. PMIDs: 14505439, 26363275), which may facilitate the investigation of pro-IL-1 α nuclear localisation in these cell lines. However, it may be most useful to investigate a myeloid cell line, which has more relevance for IL-1 α function.

We have addressed these points by adding a sentence to the introduction 'Nuclear localisation of pro-IL-1 α has been observed in human and mouse cells.' We have also added the following sentences to the discussion 'Although to our knowledge there is no current evidence regarding pro-IL-1 α nuclear localisation in cells isolated from the NLS mutant species identified in this study, cell lines have been previously derived from various cetacean and marsupial species, including toothed whales, and primary cells have also been cultured³⁰⁻³³. Thus, these cells could potentially be used to investigate pro-IL-1 α nuclear localisation and function, although myeloid cell lines from these species may offer more relevance for IL-1 α expression and function.'

Although it was not in a native non-human mammalian cell line, we believe that the experiment where we fused the Orca IL-1 α pro-domain onto the human IL-1 α mature domain provides evidence that the loss of NLS conservation identified in toothed whales would result in loss of pro-IL-1 α nuclear localisation. Although of course it would be very interesting to investigate this in primary Orca cells if they, and appropriate antibodies, were available.

6. In most mammalian species, one of the HAT binding domains (amino acid residues 7-19) is more conserved, while the other (amino acid residues 98-108) is less conserved (Fig. 6). What are the biological implications of this mutation? Does the HAT binding domain have a conserved sequence similar to the KKRR motif in the NLS of pro-IL-1 α , and does mutation of such a conserved sequence result in loss of HAT binding function?

This is an interesting comment raised by the reviewer. There is currently insufficient evidence in the literature to determine the potential biological consequences of poor conservation of the second HAT-binding domain (amino acid residues 98-108). Furthermore, we do not know whether there is a small conserved sequence within the HAT-binding domains that is critical for interaction with HAT proteins. The HAT-binding regions of pro-IL-1 α were identified through deletions of the N- and C-terminal regions of the pro-IL-1 α pro-domain that abolished IL-1 α 's interaction with EP300, without specifically identifying critical motifs within those regions (PMID: 14612453). Interestingly, deletion of either the C-terminal region of the pro-domain, or the N-terminal region, or both of them together, all reduced the functional interaction of pro-IL-1 α with EP300, suggesting that both HAT-binding domains are required for this interaction (PMID: 14612453). Thus, it is possible that poor conservation of this second HAT-binding domain could affect the interaction of pro-IL-1 α with EP300.

To address these points, we have now added the following text to the discussion: 'We observed that the second HAT-binding domain (amino acid residues 98-108) was generally more poorly conserved than the N-terminal HAT-binding domain. It is currently unclear what level of sequence or structural conservation is required to retain HAT-binding capabilities, and whether there are specific conserved motifs within the HAT-binding domain that are critical for HAT binding. It has been shown that deletion of either HAT-binding domain region alone is sufficient to reduce the functional interaction between pro-IL-1 α and EP300, suggesting that both domains are required for this interaction ¹⁰.'

Minor points:

1. There is no scale bar for the microscopy images in Figure 5.

Thank you for spotting this, we have now added scale bars to the merged images in Figure 5, and have updated the figure legend accordingly.

2. The author need to highlight proteins discovered in Figure 2C.

We have now annotated several significantly enriched proteins in the volcano plot in Figure 2C, as well as in Supplementary Figure 3C, thank you for this suggestion.

Reviewer #4 (Remarks to the Author):

This study attempted to employ TurboID-based proximity labeling to analyze the proximal interactome of Interleukin (IL)-1 α in order to understand the evolution in IL-1 α 's nuclear localization signal. Although several proteins in proximity to IL-1 α were identified through proximity labeling, there is insufficient experimental evidence to confirm whether these proteins indeed bind to IL-1 α . Additionally, the logical transition between the first part of the study, which focused on proximity labeling analysis, and the second part, which examined the evolution of the nuclear localization signal, seems somewhat disjointed, lacking a strong overall coherence.

We thank the reviewer for their feedback. In addressing the other reviewers' comments, we have taken care to improve the flow and coherence of the manuscript.

REVIEWER COMMENTS

Reviewer #1 (Remarks to the Author):

The authors have addressed all my concerns

Reviewer #2 (Remarks to the Author):

Authors Rose Wellens et al. have worked intensively on the improvements of their manuscript on "Proximity labelling and evolutionary evidence reveal insights into IL-1 α nuclear networks and function".

I have now only few minor points for their revised version:

Revised abstract:

I appreciate that some mammalian species including important marsupials were added in the abstract, even at a limited space.

Line 46 - "IL-1 β is an ancient protein":

Now the text on IL-1 β as an ancient protein is improved in the revised version and an estimated time of appearance of IL-1 β predecessor is given. This is interesting and important part. But to be consistent, can authors give at this place also the approximate time of appearance for IL-1 α for which they claim that it arose through a later gene duplication solely in mammals? It would also be interesting to reconstruct the supposed detailed ancestral sequences of all vertebrate interleukins that occurred some 425 millions of years ago according to cited reference. This is however outside of the scope of presented manuscript. It will be fine and sufficient to give here just the approximation time of appearance of both IL-1 α and IL-1 β at the same place of the revised manuscript. Some readers may wish to compare this data with several other related or seemingly unrelated proteins...

Lines 188-190 - The whole protein sequence alignment needs to be deposited in supplementary material for the overview:

It is very useful that authors have now included the full list of used sequences inclusive their accession numbers.

However, for easy orientation and search it would be better that in the first part of the table in Suppl. Data 1 all sequences are ordered alphabetically according to their latin scientific names.

Supplementary Figure 6:

OK, it is excellent that the phylogenetic tree of IL-1 α is now given in Suppl. Figure 6. However, in some nodes rather low bootstrap values were obtained. This can happen in MEGA-X but authors are

advised to remove those bootstrap values that are below 40 (at least). Anyway, the tree with the highest log likelihood from obtained dataset is presented and its topology is plausible. Indeed, for the divergence node between marsupial IL-1 α from placental mammal IL-1 α a really high bootstrap support is observed and this is important fact.

Reviewer #3 (Remarks to the Author):

The author did a great job of revising the article in depth and answering my questions completely. I suggest to publish.

Reviewer #4 (Remarks to the Author):

Well, the author has addressed my concerns.

Reviewer #1 (Remarks to the Author):

The authors have addressed all my concerns

Reviewer #2 (Remarks to the Author):

Authors Rose Wellens et al. have worked intensively on the improvements of their manuscript on "Proximity labelling and evolutionary evidence reveal insights into IL-1 α nuclear networks and function".

I have now only few minor points for their revised version:

Revised abstract:

I appreciate that some mammalian species including important marsupials were added in the abstract, even at a limited space.

Line 46 - "IL-1 β is an ancient protein":

Now the text on IL-1 β as an ancient protein is improved in the revised version and an estimated time of appearance of IL-1 β predecessor is given. This is interesting and important part. But to be consistent, can authors give at this place also the approximate time of appearance for IL-1 α for which they claim that it arose through a later gene duplication solely in mammals? It would also be interesting to reconstruct the supposed detailed ancestral sequences of all vertebrate interleukins that occurred some 425 millions of years ago according to cited reference. This is however outside of the scope of presented manuscript. It will be fine and sufficient to give here just the approximation time of appearance of both IL-1 α and IL-1 β at the same place of the revised manuscript. Some readers may wish to compare this data with several other related or seemingly unrelated proteins...

We have now modified the text to say 'IL-1 β is a protein that first arose more than 400 million years ago and is conserved throughout vertebrates, while IL-1 α arose as a gene duplication of IL-1 β between 320 and 180 million years ago in the proto-mammal, and is therefore present only in mammals ³.'

Lines 188-190 - The whole protein sequence alignment needs to be deposited in supplementary material for the overview:

It is very useful that authors have now included the full list of used sequences inclusive their accession numbers.

However, for easy orientation and search it would be better that in the first part of the table in Suppl. Data 1 all sequences are ordered alphabetically according to their latin scientific names.

We have now ordered all the species in the alignment, and in the 'species ID' tab, in Supplementary Data 1 alphabetically according to their latin names.

Supplementary Figure 6:

OK, it is excellent that the phylogenetic tree of IL-1 α is now given in Suppl. Figure 6. However, in some nodes rather low bootstrap values were obtained. This can happen in MEGA-X but authors are advised to remove those bootstrap values that are below 40 (at least). Anyway, the tree with the highest log likelihood from obtained dataset is presented and its topology is plausible. Indeed, for the divergence node between marsupial IL-1 α from placental mammal IL-1 α a really high bootstrap support is observed and this is important fact.

We have now removed any bootstrap values that were below 40, and have stated this in the figure legend.

Reviewer #3 (Remarks to the Author):

The author did a great job of revising the article in depth and answering my questions completely. I suggest to publish.

Reviewer #4 (Remarks to the Author):

Well, the author has addressed my concerns.

REVIEWERS' COMMENTS

Reviewer #2 (Remarks to the Author):

Authors Rose Wellens et al. have in the second revised version of their manuscript promptly improved my remaining minor concerns. Now, after intensive improvements I agree that this manuscript can be accepted for publication.